# Leaf Transcription Factor Family Analysis of Halophyte *Glaux maritima* under Salt Stress

**Rui Gu [1], Zhiqiang Wan [2], Fang Tang [1], Fengling Shi [1,*] and Mengjiao Yan [3]**

[1] Key Laboratory of Grassland Resources of Ministry of Education, College of Grassland, Resources and Environment, Inner Mongolia Agricultural University, Hohhot 010019, China; guruiwudan2008@163.com (R.G.); tang_lab@163.com (F.T.)

[2] College of Geographical Science, Inner Mongolia Normal University, Hohhot 010020, China; xiaoqiang1988117@126.com

[3] Plant Protection Institute, Inner Mongolia Academy of Agricultural & Animal Husbandry Sciences, Hohhot 010031, China; ymjzbs2015@163.com

* Correspondence: nmczysfl@126.com

**Abstract:** The reduction of crop yield caused by soil salinization has become a global problem. Halophytes improve saline alkali soil, and the halophyte transcription factors that regulate salt stress are crucial for improving salt tolerance. In this study, 1466 transcription factors were identified by transcriptome sequencing analysis of *Glaux maritima* leaves after salt stress (0, 600, and 800 mM/L NaCl). Their genes were distributed across 57 transcription factor families. KEGG and GO analyses showed significant enrichment in 14 pathways, with a total of 54 functions annotated. Gene expression analysis showed 820 differentially expressed genes distributed in 11 transcription factor families, including ERF, bHLH, WRKY, and NAC, and 8 expression modules. KEGG analysis revealed four genes with significant positive regulation: *ABF2* (Unigene0078257) in the ABA signaling pathway, *EIN3* (Unigene0000457 and Unigene0012139), and *EIL1* (Unigene0042139) involved in ethylene signal transduction, and two with negative regulation, *MYC1/2* (Unigene0009899 and Unigene0027167) in the main regulator of Jasmonic acid signal transduction. Protein–protein interaction networks suggested *ABF2* and *MYC1/2* as important transcription factors regulating *G. maritima* salt tolerance. Overall, the salt-tolerant transcription factors discovered in this study provide genetic resources for plant salt tolerance inheritance, and lay a theoretical foundation for the study of the salt-tolerant molecular mechanism of the halophyte *Glaux maritima*.

**Keywords:** *Glaux maritima*; transcription factors; ABFs; EIN3/EILs; MYCs; PPI





## 1. Introduction

As one of the main abiotic stress factors, salinity plays an important role in limiting crop growth, development, and yield [1,2]. In order to adapt to adverse environmental stress, a series of genes in plants are synchronously expressed, and various mechanisms are evolved to regulate plant growth and development [3,4]. Searching for candidate genes and analyzing signal mechanisms in the abiotic stress response are crucial for cultivating stress-resistant and tolerant crops [5]. Transcription factors (TFs), as important regulators of plant growth, development, and stress responses, often participate in different and complex signal networks [6,7].

With recent in-depth and extensive research on the mechanism of the plant response to salt stress, a large number of involved TF families, including NAC (NAM, ATAF1/2, CUC2), ERF/AP2 (Ethylene Response factor/APETALA2), bZIP (basic leucine-zipper), bHLH (Basic Helix-Loop-Helix), MYB (v-myb avian mye1oblastosis viral oncogene homolog), and WRKY, have been discovered [8]. Although the main transcriptional regulator of salt stress is not yet clear, a large number of studies have shown that TFs related to the regulation of ABA, jasmonate, ethylene, and other hormones play important roles in

the salt stress response. ABA mainly mediates transcriptional regulation through the AREB/ABF subfamily of bZIP TFs [9,10]. For example, *ABF2* overexpression increases the resistance of plants to salt stress [11]. Overexpression of the ABA-responsive TFs dynamic influencer of gene expression (DIG) and DIG-like (DIL) leads to hypersensitivity to high-salt conditions or ABA [12]. MYC2, as the master TF of jasmonate signaling, plays a positive role in regulating plant salt tolerance [13,14]. EIN3, as a TF that mediates the core ethylene signaling pathway, improves plant salt tolerance through the expression of the DELLA protein after salt treatment [15,16]. *EIN3* directly binds to a salt response gene and activates its expression through two downstream TFs, ethylene response factor 1 (ERF1) and ethylene and salt inducible 1 (ESE1), to improve plant salt tolerance [17,18]. In *Arabidopsis thaliana*, the nuclear calcium-binding gene *AtbHLH28* (*ATNIG1*) is believed to play an important role in the regulation of plant salt stress signals by binding to the E-box motif (CANNTG) of salt-stress-related gene promoters [19]. Overexpression of *bHLH106* in different plants increases tolerance to NaCl [4], and co-expression of *bHLH17* and *WRKY28* leads to resistance in *A. thaliana* to various abiotic stresses [20]. Salt has a significant inductive effect on wheat *TaWRKY93*, whose overexpression improves salt tolerance in *A. thaliana*. This indicates that a single *WRKY* gene can influence the complex process of salt tolerance in plants [21]. Although the specific mechanism of this TF is still unclear, its role in salt tolerance has generally been verified in *Arabidopsis* or the original species through transgenic methods [7]. At the same time, many other TFs are also involved in regulating gene expression induced by salt. High-throughput sequencing technology is helpful in identifying such TFs involved in the salt stress response [8].

Halophytes are important plant resources. Researchers mainly study their phenotype, physiology, and molecular biology to explore their salt tolerance characteristics and mechanisms and to provide theoretical bases for improving plant salt tolerance [22–26]. After analyzing the whole genome of *Limonium bicolor*, Yuan et al. [27] found the important genes affecting the formation of salt glands and the salt tolerance adaptation mechanism. Zhao et al. [28] analyzed transcriptome-level data of *Arachis paniculata* after salt stress and found that ethylene response factor (ERF), MYB, WRKY, and bZIP family TFs played important roles in plant salt tolerance.

As an important agricultural and animal husbandry area in northern China, Inner Mongolia has been experiencing an annual increase in saline alkali land area due to its arid climate and the fact that irrigation is the main planting method. Therefore, there is an urgent need to study salt-tolerant mesophytes that are suitable for growing in the ecological environment of the region in order to provide genetic resources for cultivating salt-tolerant and highly resistant agricultural and pastoral crops. Based on the previous analysis of the morphology, physiology, and transcriptome of *Glaux maritima*, this study analyzed the salt-tolerance-related TFs of the salt-secreting halophyte *Glaux maritima* at the transcription level and identified six important genes distributed in the EIN, bHLH, and bZIP TF families. Clustering and expression analyses of these genes revealed that they play key roles in the response to high-salt conditions and could provide valuable genetic resources for cultivating salt-tolerant crops.

## 2. Materials and Methods

Wild *G. maritima* seedlings were cultivated with 1/2 Hoagland nutrient solution for 15 days in the Key Laboratory of the Ministry of Education, College of Grassland, Resources, and Environment, Inner Mongolia Agricultural University (wild *G. maritima* seedlings were collected from the Hailiutu base of Inner Mongolia Agricultural University Science and Technology Park, Bikeqi Town, Tumet Left Banner, Hohhot, Inner Mongolia, N: 40°38′, E: 111°28′, Height: 1060 m.). The optimal growing conditions included a steady 25 °C temperature, 60% relative humidity, and 1600 lx of light. Selected *G. maritima* seedlings were subjected to NaCl concentrations of 0 (N0), 600 (N3), and 800 (N4) mM for 24 h [29,30]. (Attached Figure S1) Moreover, leaf samples of *G. maritima* were collected in triplicate

for molecular analysis, directly frozen in liquid nitrogen, and kept at −80 °C for RNA isolation. [31].

## 2.1. RNA Isolation

Total RNA was extracted using a TRIzol reagent kit (Invitrogen, Carlsbad, CA, USA), as per the manufacturer's instructions [31]. To verify the integrity of the RNA, we processed it through RNase-free agarose gel electrophoresis and then analyzed the results using an Agilent 2100 Bioanalyzer (Agilent Technologies, Palo Alto, CA, USA). Following total RNA isolation, Dynabeads$^{TM}$ Oligo(dT)$_{25}$ beads (Thermo Fisher Scientific, Waltham, MA, USA) were used to remove rRNA from eukaryotic mRNA, and a Ribo-Zero$^{TM}$ Magnetic Kit was used to do the same for bacterial mRNA (Illumina, San Diego, CA, USA) (Attached Figure S2, Supplementary Table S1). The enriched mRNA was fragmented using a fragmentation buffer before being used for reverse transcription into cDNA. Second-strand cDNA was synthesized using DNA polymerase I, deoxynucleoside triphosphates, RNase H, and a buffer. Furthermore, cDNA extraction was carried out using a QiaQuick PCR extraction kit (Qiagen, Venlo, The Netherlands). The cDNA products were end-repaired, had an extra base added, and ligated using Illumina sequencing adapters. Finally, agarose gel electrophoresis was used to separate the ligation products by size, which were then amplified by PCR and sequenced on an Illumina Novaseq 6000 (Illumina).

## 2.2. qRT-PCR Verification

TransScript One-Step gDNA Removal and cDNA Synthesis SuperMix (CAS:AT311; Transgen Biotech Co., Ltd., Beijing, China) was used to reverse transcribe RNA. PerfectStart$^{TM}$ Green qPCR SuperMix (CAS:AQ601, Transgen) containing 50 ng cDNA template and $50\times$ passive reference dye for a final volume of 20 μL was used for qRT-PCR on each sample in a LightCycler 480 (Roche, Basel, Switzerland). The amplification protocol was as follows: 94 °C for 30 s, 45 cycles at 94 °C for 5 s, 60 °C for 34 s, and 72 °C for 10 s, and then melting curve analysis was carried out at 95 °C 15 s, 60 °C for 1 min, and 95 °C for 15 s. Supplementary Table S4 lists the sequences of the specific primers used for qRT-PCR. β-Actin was used as an internal control. All qRT-PCR analyses were performed in three replicates.

## 2.3. Data Analysis

Excel 2019 was used for preliminary data processing. Several variables were analyzed using univariate analysis in SPSS (version 19.0; IBM, Armonk, NY, USA) to demonstrate their importance across the range of salt-stress treatments, and the differentially expressed gene (DEG) statistics were based on the results of differential analysis. The screening of significantly DEGs was based on $|\log_2$ fold change$| > 1$ and a false discovery rate $< 0.05$. The expression patterns of the genes were clustered using trend analysis, followed by analysis using the Short Time-series Expression Miner (STEM v1.3.13; http://www.cs.cmu.edu/~jernst/stem accessed on 22 July 2022), and enrichment analysis was carried out using Gene Ontology (GO) and Kyoto Encyclopedia of Genes and Genomes (KEGG) resources. To obtain enriched GO keywords and pathways, we used a threshold of Q $\leq 0.05$. Relative gene expression levels were calculated using the $2^{-\Delta\Delta Ct}$ method (Livak, 2001) and analyzed via analysis of variance (ANOVA) with a significance threshold of $p < 0.05$. The DEGs of the protein–protein interaction (PPI) network were predicted by STRING [32] (https://string-db.org/ accessed on 22 October 2022), and the drawn network diagram and network node degree were analyzed by Cytoscape 3.5.1. TFs were identified using Plant-TFDB 4.0 (http://planttfdb.cbi.pku.edu.cn/ accessed on 1 October 2022).

## 3. Results

### 3.1. Overall Analysis of Transcription Factors of Glaux Maritima under Salt Stress

Through assembly analysis and comparison of transcriptome data from 9 cDNA libraries of *Glaux maritima*, 1466 TF genes were counted and distributed across 57 TF

families, mainly the ERF, bHLH, MYB-related, NAC, and WRKY families. The TF families with the largest numbers of genes were ERF (133, 9.07%), bHLH (123, 8.39%), MYB-related (95, 6.48%), NAC (94, 6.41%), WRKY (84, 5.73%), $C_2H_2$ (81, 5.53%), bZIP (70, 4.77%), and MYB (66, 4.50%) (Figure 1).

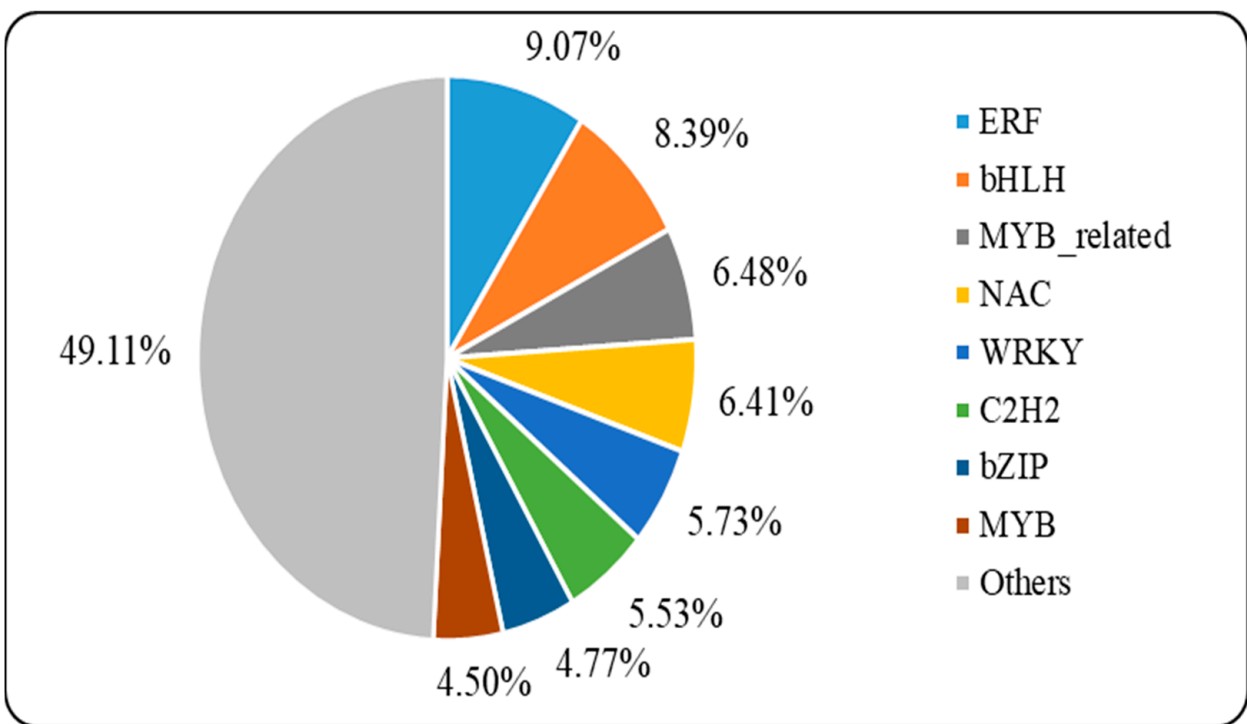

**Figure 1.** Pie chart showing the percent share of identified transcription factors in each Transcription factor family. Note: The figure shows the distribution of 1466 TF genes in the top 8 TF families in the *Glaux maritima*. The 8 TF families were ERF (133, 9.07%), bHLH (123, 8.39%), MYB related (95, 6.48%), NAC (94, 6.41%), WRKY (84, 5.73%), $C_2H_2$ (81, 5.53%), bZIP (70, 4.77%), and MYB (66, 4.50%).

*3.2. KEGG and GO Enrichment Analysis of All Transcription Factors*

In order to better understand the main biochemical and signal transduction pathways related to the obtained TFs, KEGG pathway analysis was conducted on the TF family (Figure 2). A total of 95 TF genes were significantly enriched in 14 pathways, including plant hormone signal transduction (66 unigenes), the MAPK (Mitogen-activated protein kinases) signaling pathway (34 unigene), plant–pathogen interaction (18 unigene), circadian rhythm (5 unigene), base exception repair (1 unigene), and DNA replication (1 unigene). These annotations provide important information for studying the specific functions, processes, and mechanisms involved in the salt tolerance of halophytes.

To understand the salt tolerance characteristics of the TFs, GO annotation and enrichment analysis were carried out on all TFs after salt treatment (Figure 3). A total of 54 functions were annotated, and 1191 genes were enriched in these processes. The highly enriched molecular functions were binding (GO:0005488) and nucleic acid binding transcription factor activity (GO:0001071), and the highly enriched cell components were cell (GO:0005623), cell part (GO:0044464), and organelle (GO:0043226). In addition, biological processes such as the cellular process (GO:0009987), the metabolic process (GO:008152), the regulation of biological process (GO:0050789), and the biological regulation (GO:0065007) were highly enriched at all stages.

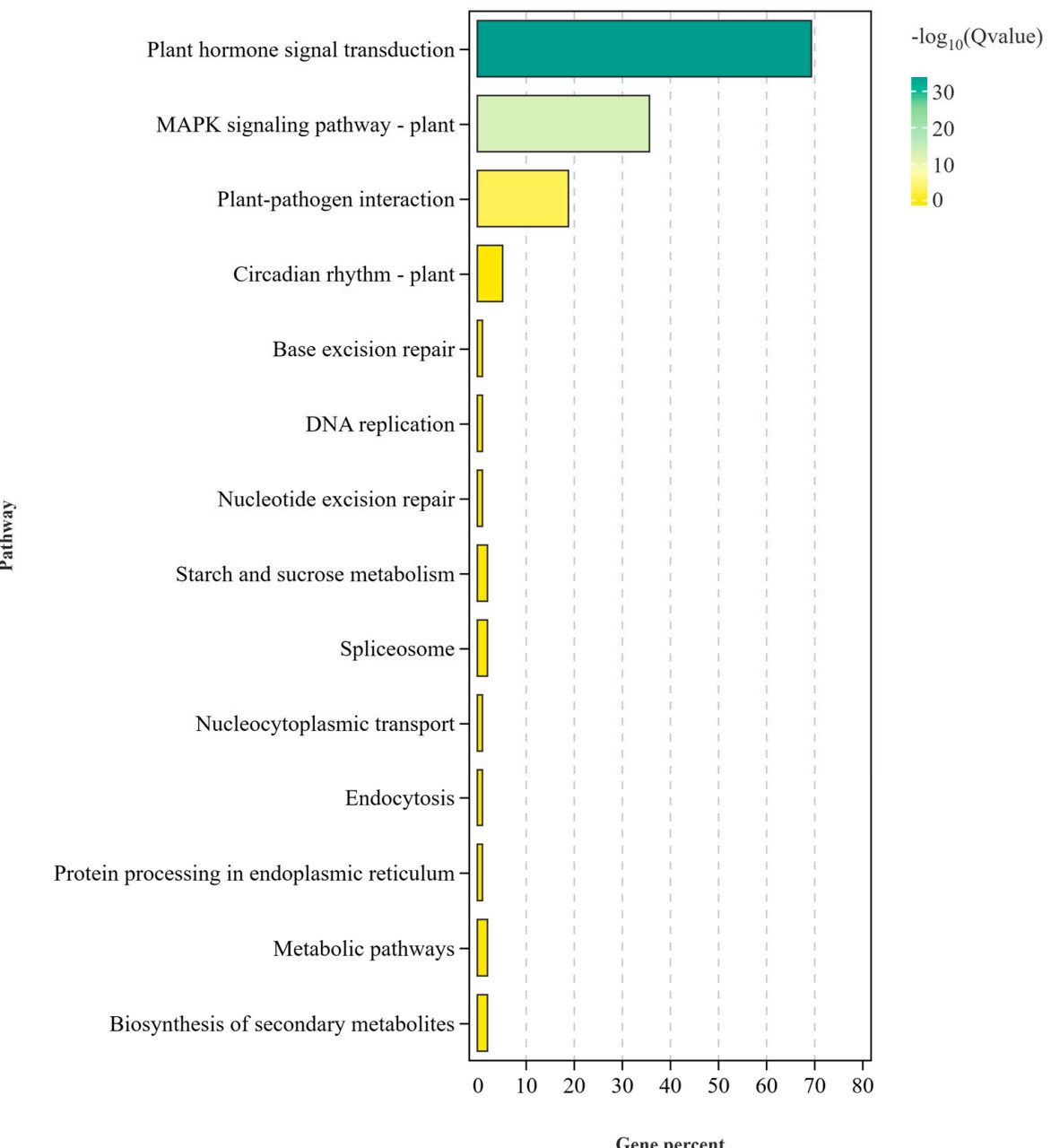

**Figure 2.** KEGG pathway enrichment analysis of the transcription factor genes. Note: under salt stress, a total of 95 TF genes were successfully annotated in the KEGG database for *Glaux maritima* TF genes. The KEGG pathways with the highest number of annotated genes were Plant hormone signaling pathway (66), MAPK signaling pathway-plant (34) and plant pathogen interaction (18).

### 3.3. Trends in Transcription Factor Gene Expression under Different Levels of Salt Stress

All TF genes were subjected to trend analysis, and 820 genes with significant differences in expression were identified. Differential gene expression analysis showed that these genes were distributed in eight expression modules (Figure 4). The genes with significantly different expression levels after different salt treatments ($p < 0.05$) were all distributed in the four modules of Profile 0, 1, 6, and 7, with 83, 239, 222, and 64 genes, respectively. These DEGs were mainly distributed in the ERF (96, 11.71%), bHLH (75, 9.15%), WRKY (67, 8.17%), NAC (65, 7.93%), MYB-related (49, 5.98%), $C_2H_2$ (47, 5.73%), MYB (45, 5.49%), bZIP (37, 4.51%), C3H (25, 3.05%), GRAS (25, 3.05%), and HD-ZIP (22, 2.68%) families

(Figure 5). Comparing the numbers of DEGs in all TF families showed that most DEGs were concentrated in the top ten TF families mentioned above (Figure 6).

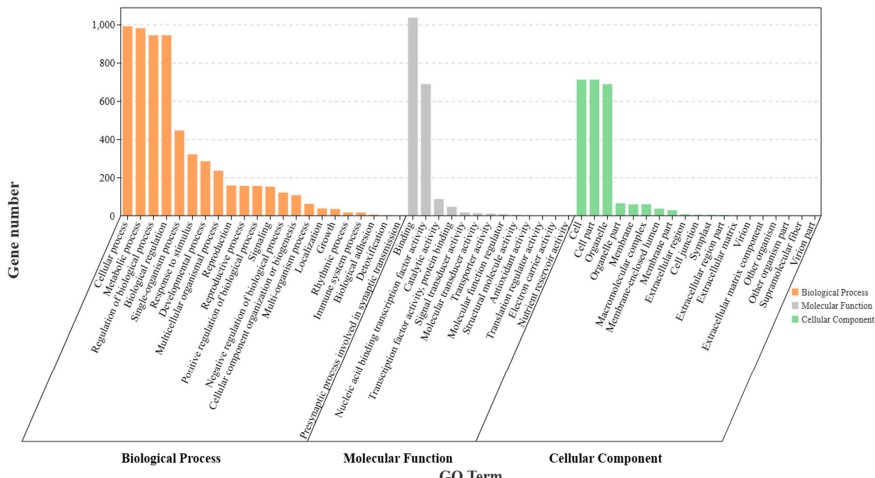

**Figure 3.** GO enrichment analysis of the transcription factor genes. Note: under salt stress, a total of 54 functions were anno-tated, and 1191 genes were enriched in GO annotation and enrichment analysis. Among the three ontological processes, binding (GO:0005488) and nuclear acid binding transcription factor activity (GO:0001071) had the highest distribution of Molecular Func-tion; (GO:0005623), cell part (GO:0044464), and organelle (GO:0043226) had the highest distribution of Cell Component; cellular process (GO:0009987), metabolic process (GO:008152), regulation of biological process (GO:0050789), and biological regulation (GO:0065007) had the highest distribution of Biological Process.

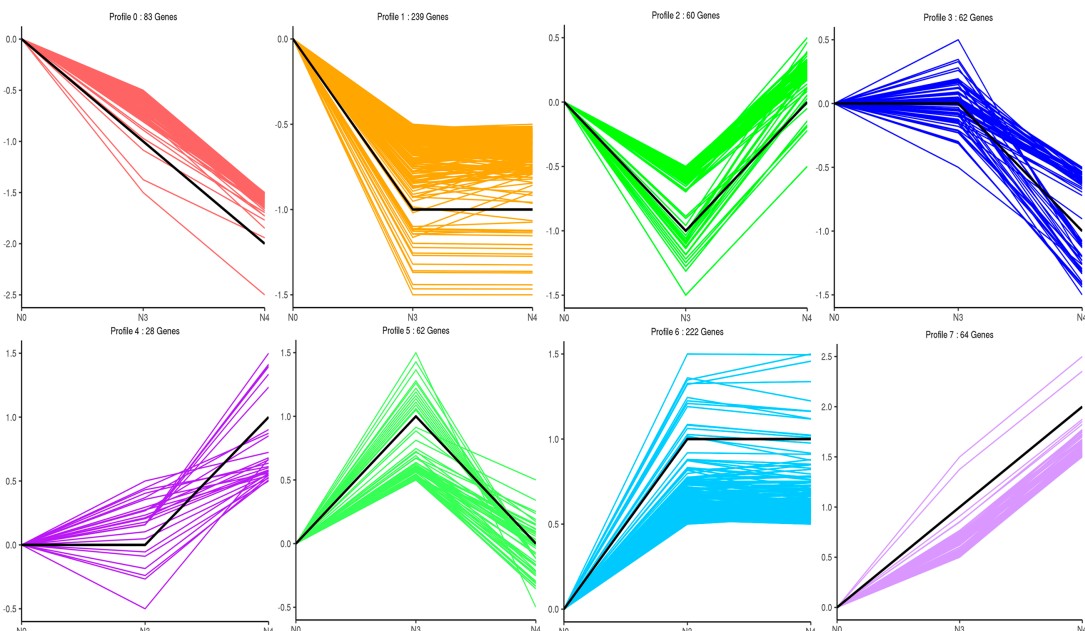

**Figure 4.** Gene expression module diagram for trend analysis of all transcription factors. Note: After trend analysis, there were a total of 820 differentially expressed genes distributed in 8 expression modules. They were profile 0 (83 genes), profile 1 (239 genes), profile 2 (60 genes), profile 3 (62 genes), profile 4 (28 genes), profile 5 (62 genes), profile 6 (222 genes), and profile 7 (64 genes). In addition, the gene expression in the four modules of profile 0, 1, 6, and 7 showed significant differences after different concentrations of salt stress ($p < 0.05$). N0: 0 mM/L NaCl stress; N3: 600 mM/L NaCl stress; N4: 800 mM/L NaCl stress.

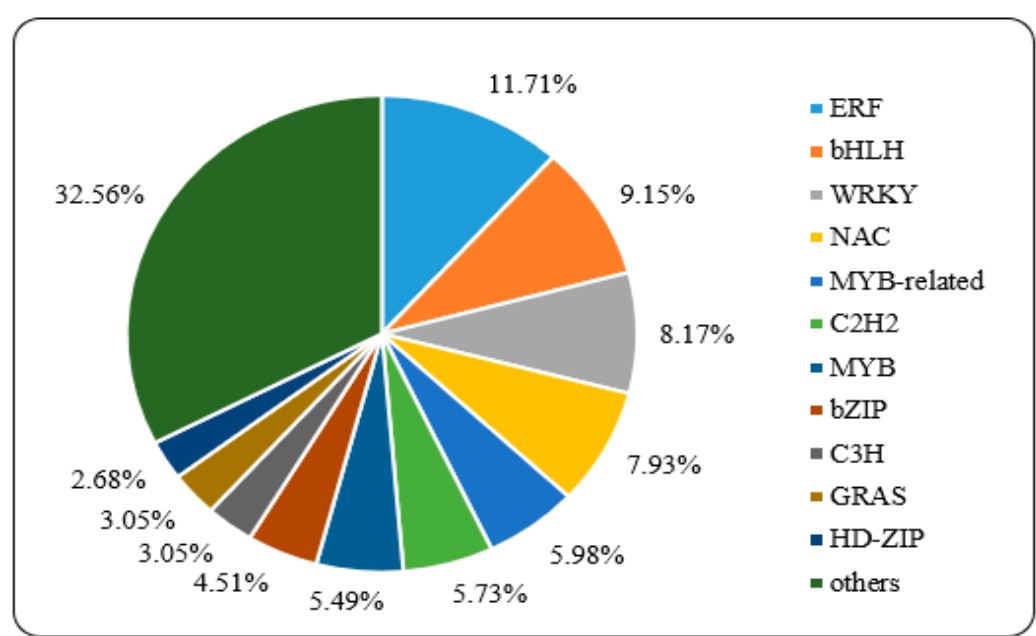

**Figure 5.** Differentially expressed transcription factor gene statistics. Note: The figure shows the distribution of 820 TF genes in the top 11 TF families in the *Glaux maritima*. The 11 TF families were ERF (96, 11.71%), bHLH (75, 9.15%), WRKY (67, 8.17%), NAC (65, 7.93%), MYB-related (49, 5.98%), $C_2H_2$ (47, 5.73%), MYB (45, 5.49%), bZIP (37, 4.51%), C3H (25, 3.05%), GRAS (25, 3.05%), and HD-ZIP (22, 2.68%).

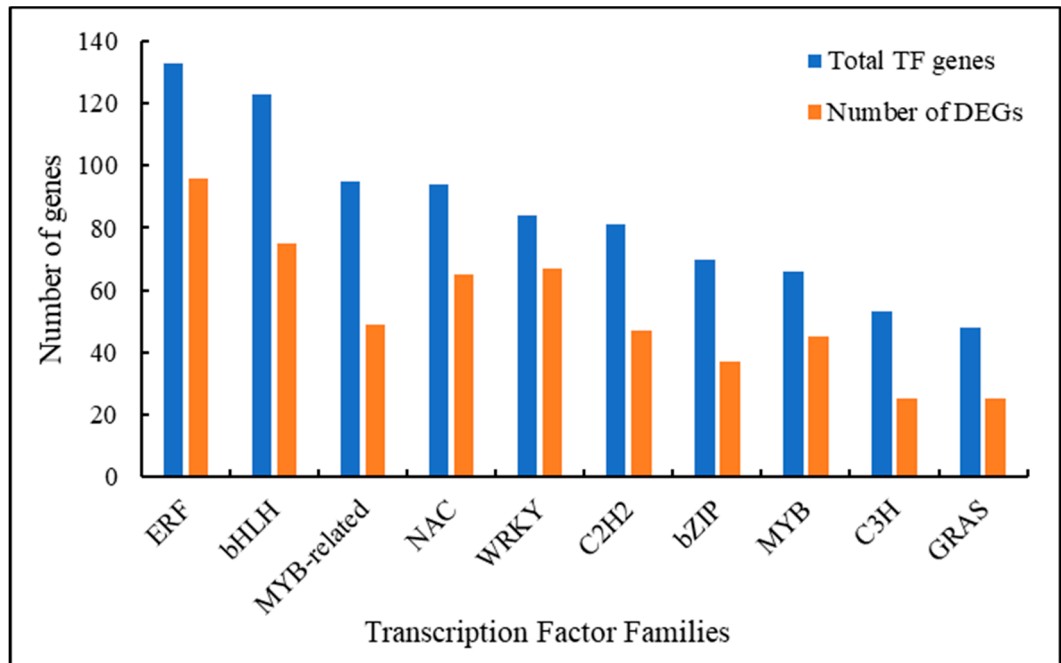

**Figure 6.** Numbers of genes in the top 10 transcription factor families. Note: Statistics on the total number of TF in the top 10 TF families of *Glaux maritima* and the number of differentially expressed genes in trend analysis. The 10 TF families were ERF (TF: 133 genes, DEGs: 96 genes), bHLH (TF: 123 genes, DEGs: 75 genes), MYB-related (TF: 95 genes, DEGs: 49 genes), NAC (TF: 94 genes, DEGs: 65 genes), WRKY (TF: 84 genes, DEGs: 67 genes), $C_2H_2$ (TF: 81 genes, DEGs: 47 genes), bZIP (TF: 70 genes, DEGs: 37 genes), MYB (TF: 66 genes, DEGs: 45 genes), C3H (TF: 53 genes, DEGs: 25 genes) and GRAS (TF:48 genes, DEGs: 25 genes). TF: transcription factor, DEGs: differentially expressed genes.

### 3.4. Differentially Expressed Transcription Factors under Salt Stress

The differentially expressed TF genes were identified. Under N3 and N4 salt concentration stress, 33 up-regulated and 35 down-regulated DEGs were identified (Figure 7a,c). Differentially expressed TFs included ERF, WRKY, EIN, MYC, MYB, ABF, ARF, and NAC family members. To further understand the expression trends of these single genes, a trend change diagram (Figure 7b,d) was constructed from the log$_2$ FPKM values of these 68 genes. This comprehensive analysis showed that under N3 and N4 conditions, the expression of all single genes increased or decreased in a consistent trend.

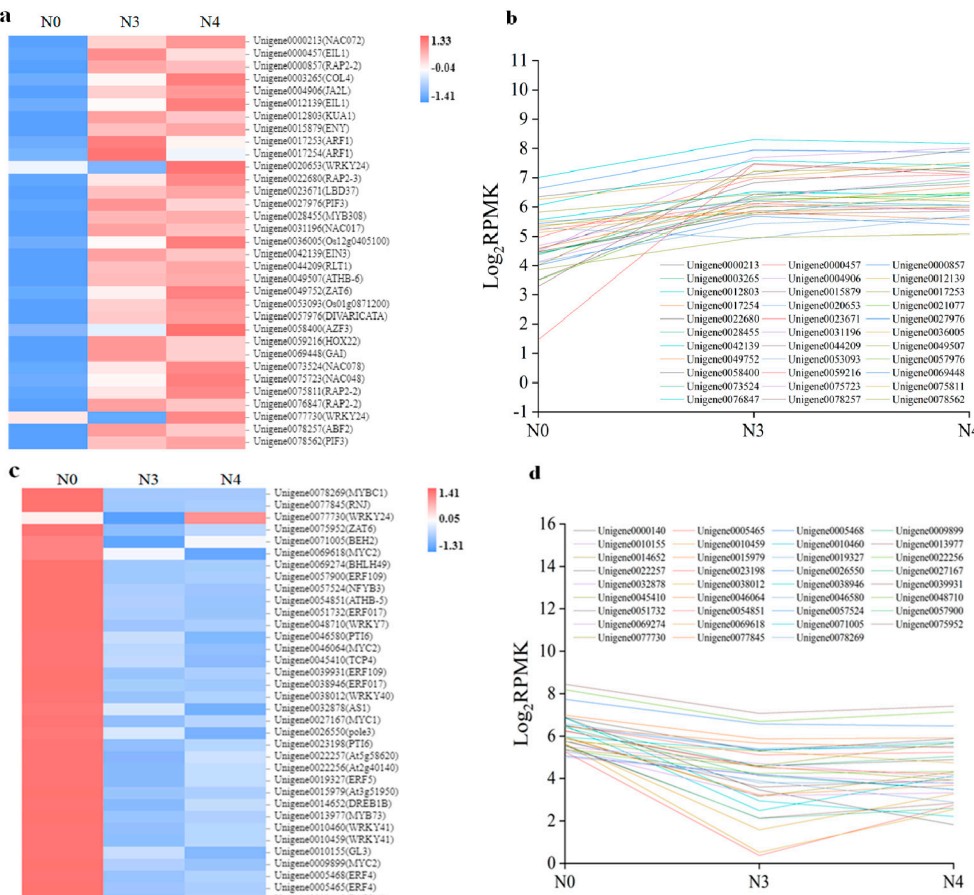

**Figure 7.** Heat map of differential transcription factor gene expression and log$_2$ RPKM expression trends. Note: (**a**): Heat map of 33 up-regulated differentially expressed genes under different salt concentrations; (**b**): log$_2$ RPKM expression trends of 33 upregulated differentially expressed genes under different salt concentrations; (**c**): Heat map of 35 down-regulated differentially expressed genes under different salt concentrations; (**d**): log$_2$ RPKM expression trends of 35 down-regulated differentially expressed genes under different salt concentrations. N0: 0 mM/L NaCl stress; N3: 600 mM/L NaCl stress; N4: 800 mM/L NaCl stress.3.5. Analysis of ABF, EIN, and MYC transcription factors involved in the salt stress response of *Glaux maritima*.

### 3.5. Analysis of ABF, EIN, and MYC Transcription Factors Involved in the Salt Stress Response of Glaux Maritima

The plant hormone ABA activates a series of signaling pathways to activate the resistance mechanisms of plants under abiotic stress. ABF TFs play an important role in the regulation of the ABA signal pathway. They specifically recognize the AREB (ABA responsive element binding protein) class of basic leucine zipper proteins and are mainly involved in the regulation of ABA and stress. Positive regulatory ABF TFs related to salt stress have been found in *Arabidopsis*, tobacco, sweet potato, rape, and other crops [9,27,33,34]. In this study, five DEGs related to ABF TFs (*Unigene0003302, Unigene0007479, Unigene0007480,*

*Unigene0007482*, and *Unigene0078257*) were found after salt stress in *Glaux maritima*. *Unigene0078257* expression levels significantly increased with the increase in NaCl concentration, while the expression levels of the other four genes showed no significant changes. In order to better understand the relationship between these TFs and salt tolerance, a phylogenetic tree of ABF TFs was constructed using ABF protein sequences downloaded from the Plant TF Database (Figure 8a). Phylogenetic analysis showed that the five genes were listed in two branches. *Unigene0007479*, *Unigene0007480*, and *Unigene0007482* were in the same branch as ABF TF proteins in model plants *A. thaliana* (AT4G34000.1 and AT4G34000.2) and *Nicotiana tabacum* (XP016458525.1 and XP016451400.1). Meanwhile, *Unigene0003302* and *Unigene0078257* had high homology with the ABF TF protein in *Actinidia chinensis* Planch (Achn026271 and Achn131571). These results provide a good theoretical basis for the cloning of and functional research on the ABF TFs of *Glaux maritima*.

bHLH, as the second largest eukaryotic TF family after MYB, participates in the defensive responses of plants and helps resist damage when plants are subjected to abiotic stress. After MYC2 was first found and its function identified in *A. thaliana*, a large number of studies showed that this TF participated in various stress responses and that it was also the main regulator of jasmonate signaling [35]. In this study, seven regulatory DEGs related to the TFs MYC1 and MYC2 (*Unigene0006978*, *Unigene0009899*, *Unigene0027154*, *Unigene0027167*, *Unigene0046064*, *Unigene0069618*, and *Unigene0077023*) were also found in *G. maritima* after salt stress. Other than *Unigene0077023* and *Unigene0006978*, whose expression levels did not change under the different concentrations of salt stress, the genes were all negatively regulated. Among them, *Unigene0009899* and *Unigene0027167* expression levels decreased significantly as salt concentration increased, while the expression levels of other genes did not significantly change under salt stress. A phylogenetic tree analysis (Figure 8b) showed that the seven genes belonged to two different branches. *Unigene0009899* and *Unigene0027154*, both related to MYC2 TFs, were in the same branch as MYC TFs in *N. tabacum* (XP016462596.1, XP016508165.1, and XP016449248.1) and *A. chinensis* (Achn31034.1). The other five genes were associated with *Gossypium hirsutum* (Gh_D12G2074), *N. tabacum* (XP016441449.1 and XP016499262.1), *Capsella grandiflora* (Cagra. 3356s0080.1. p), *A. thaliana* (AT5G46830.1, AT4G17880.1, and AT5G46760.1), and *Arabidopsis lyrate* (493087). Additionally, the MYC TF family genes in *Brassica napus* (GSBRNA2T00131278001 and GSBRNA2T00080269001) and *Brassica oleracea* (XP_013605597.1) are in the same branch. These results expand our understanding of the function of MYC TF family genes in salt tolerance and lay the foundation for further research.

In this study, nine regulatory genes related to EIN3/EIL TFs were found in the TF analysis of *G. maritima*. Three genes (*Unigene0000457*, *Unigene0012139*, and *Unigene0036810*) were related to EIL1/3 TFs, and six (*Unigene0036956*, *Unigene0036957*, *Unigene0042139*, *Unigene0071273*, *Unigene0071274*, and *Unigene0071276*) to *EIN3* TFs. Phylogenetic tree analysis showed that *Unigene0036957*, *Unigene0000457*, and the other seven genes belong to three different branches. *Unigene0000457* and genes found in *Citrus clementina* (Ciclev10000607m), *A. thaliana* (AT2G27050.1) and *Nicotiana sylvestris* (XP_009801714.1) belong to the same branch, while *Unigene0036810*, *Unigene0036956*, *Unigene0042139*, *Unigene0071273*, *Unigene0071274*, *Unigene0071276*, and genes found in *A. chinensis* (Achn347321 and Achn018431), *A. thaliana* (AT3G20770.1), *Carica papaya* (evm. model. supercontig_103.60), and *Citrus sinensis* (orange1.1g007186m) were in adjacent branches. *Unigene0000457*, *Unigene0012139*, and *Unigene0042139* expression levels were significantly up-regulated in response to salt stress, but no other genes of the EIL TF family showed significant expression changes.

### 3.6. Protein–Protein Interaction Network and Topological Analysis in Glaux Maritima

To better understand how DEGs in these TF families participate in protein regulation and mediation, we constructed a protein–protein interaction (PPI) network of the DEGs in the ABF, MYC, and EIN TF families. The results showed that the DEG related to the ABF family (*Unigene0078257*) interacted with 106 genes, forming a total of 472 interactions (Figure 9a). The DEGs related to the MYC family (*Unigene0009899* and *Uni-*

gene0027167) interacted with 349 and 348 genes, respectively (Figure 9b,c), to form 4531 and 3526 interactions, respectively, but there were no interactions with the DEGs related to the EIL family.

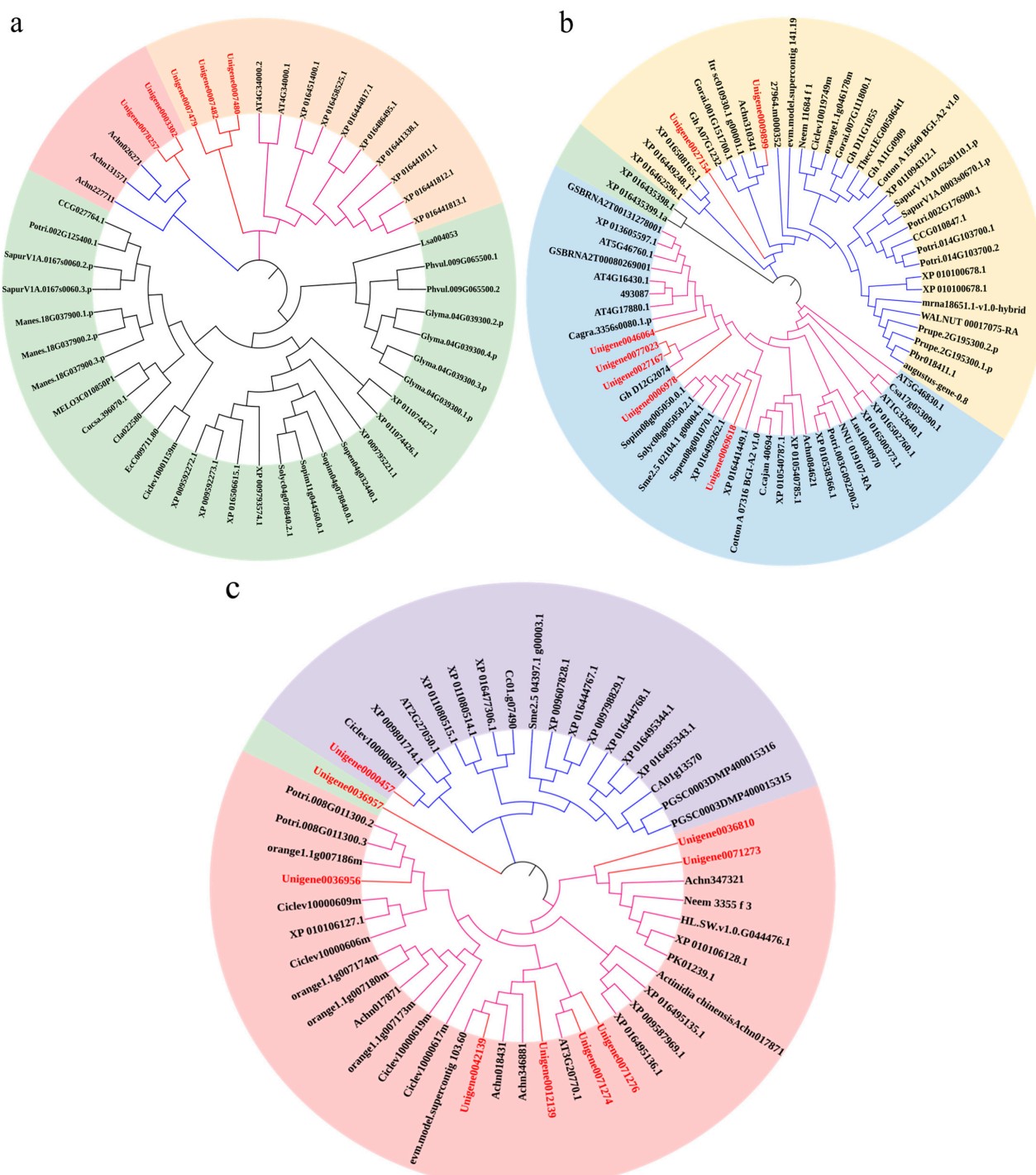

**Figure 8.** Phylogenetic tree analysis of genes in the ABF, MYC, and EIN3/EIL transcription factor families. Note: The genes were identified via trend analysis of TFs in *Glaux maritima* and are related to plant hormone signal transduction. (**a**): Phylogenetic tree analysis of ABF2 TFs. (**b**): Phylogenetic tree analysis of MYC1 and MYC2 TFs. (**c**): Phylogenetic tree analysis of EIN3 TFs.

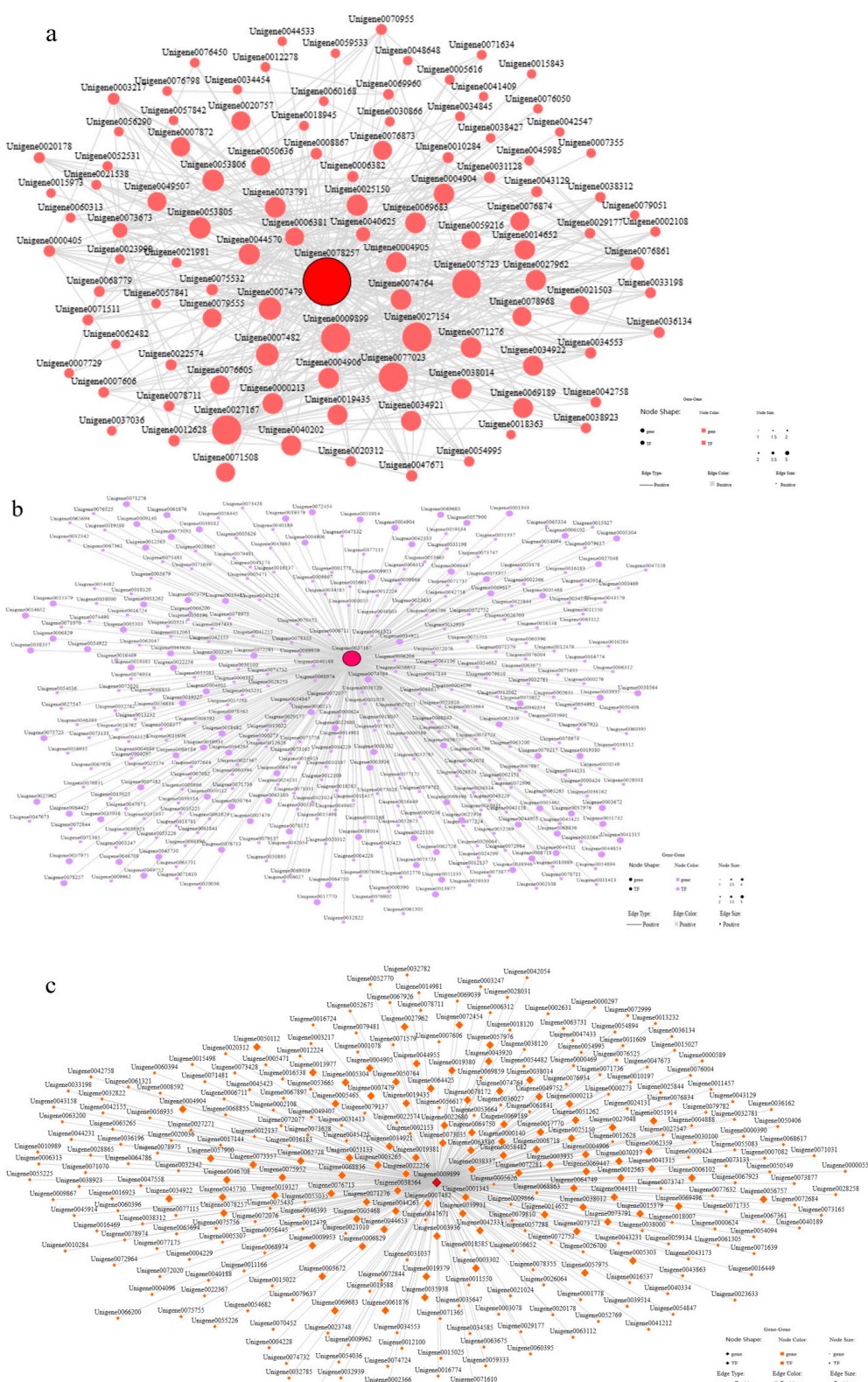

**Figure 9.** PPI network analysis on DEGs found in ABFs and MYC transcription factor families. Note: (**a**): The ABFs TF related differential gene (*Unigene0078257*) interacted with 106 genes, forming a total of 472 interactions; (**b**): The MYC TF related differential gene (*Unigene0009899*) interacted with 349 genes, forming a total of 4531 interactions; (**c**): The MYC TF related differential gene (*Unigene0027167*) interacted with 348 genes, forming a total of 3526 interactions.

### 3.7. KEGG Pathway Analysis of Differentially Expressed Genes

KEGG analysis showed that salt stress significantly increased the expression of the ABF TF family gene *Unigene0078257* downstream of SnRK2 in the ABA signaling pathway, and *Unigene0078257* expression more than doubled under N3 and N4 conditions (Figure 10). In the MAPK signaling pathway, the three EIN3/EIL TF family DEGs related to ethylene signal transduction were all up-regulated. Compared with levels after control treatment, expression levels increased more than 1.1 times after N3 and N4 treatments. In the Jasmonate signaling pathway, the two DEGs related to the MYC2 family were down-regulated, with expression levels less than 0.4 times those observed after control treatment (Figure 11).

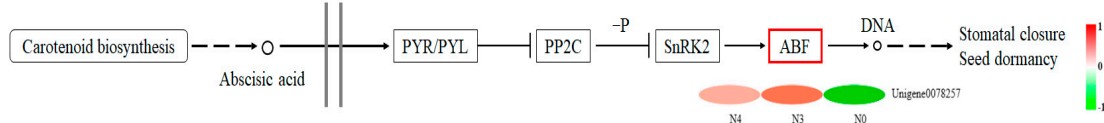

**Figure 10.** Differentially expressed genes involved in plant hormone signal transduction. Note: N0: 0 mM/L NaCl stress; N3: 600 mM/L NaCl stress; N4: 800 mM/L NaCl stress.

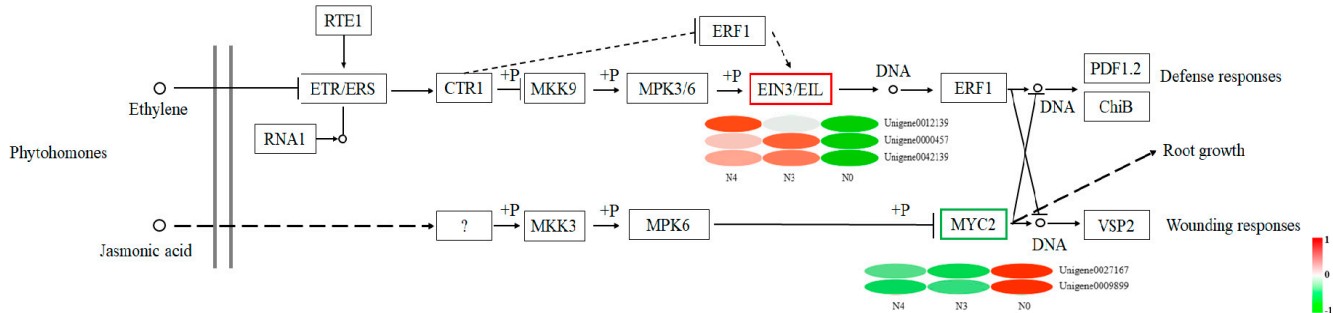

**Figure 11.** Differentially expressed genes enriched in the MAPK signaling pathway-plant. Note: N0: 0 mM/L NaCl stress; N3: 600 mM/L NaCl stress; N4: 800 mM/L NaCl stress.

## 4. Discussion

Transcriptional regulation plays an important role in plant resistance to stress. Most regulation will be realized through interactions between specific TFs and corresponding cis-acting elements. TFs, as DNA-binding proteins that specifically interact with *cis*-acting elements in the gene promoter region, will activate or inhibit transcription on their own or with other proteins so that genes are expressed in specific places at specific times. Many different TF families related to salt stress, including WRKY, avian myeloblastosis viral oncogene homolog (MYB), NAM, ATAF1/2, and CUC2 (NAC), basic leucine zipper (bZIP), APETALA2/ethylene response factor (AP2/ERF), and basic helix loop helix (bHLH), among others, have been identified by analyzing the whole genomes of different plants [36]. With the increasing global saline-alkali land area, halophytes are receiving increasing attention from researchers. Previous studies on the salt tolerance characteristics and molecular mechanisms of halophytes have been conducted at phenotypic, physiological, and molecular levels [22–26]. Halophytes could provide better genetic resources for improving plants salt tolerance. At present, the salt-tolerant TF families found in halophytes mainly include AP2/EREBP, MYB, WRKY, NAC, HD ZIP, zinc finger domain (ZnF), G-box binding proteins, 14-3-3 protein analogs, MADS box proteins, and bHLH. These TFs show different expression patterns under salt stress and have specific or universal response mechanisms for certain times and locations. Additionally, they participate in complex regulatory networks in plants [37–42]. This is similar to the results of the study on the salt-tolerant TF family in the halophytic plant *G. maritima*. This study also found that TFs in the ERF, bHLH, and bZIP TF families play a crucial role in the adaptation of *G. maritima* to high-salt environments. In this study, by combining KEGG and GO enrichment analysis and trend analysis, we analyzed the DEGs in these TF families and found that

six salt-tolerant key genes in the EIN, bZIP, and bHLH TF families were expressed in the Plant hormone signal pathway. These were similar to the research results on the gene expression regulation of ERF, MYB, bZIP, and bHLH TF families in *Arabidopsis*, *tobacco*, *soybeans*, *Medicago truncatula*, and *Zea mays* to enhance plant salt tolerance [43–46]. The molecular mechanism of salt tolerance in the halophyte *G. maritima* may be directly related to the regulation of these TFs.

Phytohormones are important endogenous molecules that regulate plant development and salt tolerance and play an important role in plants subjected to salt, drought, temperature, and other stressors. Therefore, the genes involved in the regulation of plant salt tolerance and in plant hormone signal transduction have attracted the attention of researchers. In attempts to understand and improve plant salt tolerance, researchers have revealed and verified a large number of positive regulatory genes in the ABA and ethylene-related pathways [47,48]. Using an immunoblotting experiment, Zhu et al. [49] found that *AtCDPK4* and *AtCDPK11* in *Arabidopsis* can phosphorylate *AtABF1* and *AtABF4* and improve tolerance to high-salt and drought stress by regulating stomatal movement. Liu et al. [50] found that under salt stress, the chlorophyll content of *A. thaliana abf3* and *abf4* mutants decreased, and the yellowing rate increased after salt stress, indicating that the *AtABF3* and *AtABF4* genes played key roles in the salt-stress response. When subjected to salt stress, transgenic potatoes overexpressing *AtABF4* exhibited higher relative water content and lower stomatal conductance and transpiration rates than levels in wild-type potatoes, hallmarks of stronger salt resistance [51]. Transgenic *A. thaliana* overexpressing the *Ipomoea batatas* (L.) *IbABF4* gene showed a salt-resistant phenotype with higher photosynthetic efficiency and lower malondialdehyde and hydrogen peroxide content than levels in wild-type plants [52]. *BnaABF2* overexpression in *Arabidopsis* showed that transgenic plant leaves can improve salt tolerance by reducing stomatal aperture size and inhibiting water loss after salt stress [53]. In this study, we found that five ABF TF family genes involved in the ABA signal pathway may be key regulators of the *G. maritima* salt stress response. Phylogenetic tree analysis of the genes showed similarity to ABF TF protein sequences found in plants such as *A. thaliana*, *N. tabacum*, and *A. chinensis*. Additionally, PPI network analysis of Unigene0078257, a DEG that was up-regulated by salt stress, showed connections with 106 genes to form 472 interactions (Figure 9a). This result suggests that this gene plays a key role in regulating the salt tolerance of *G. maritima* and represents an excellent genetic resource for the breeding of salt-resistant crops.

As the first discovered gaseous plant hormone, ethylene is involved in the complete processes of plant growth and development and in various stress responses. The ethylene signaling pathway has also been clearly described. Ethylene insensitive 3/ethylene insensitive 3-like (EIN3/EIL) is a family of small TFs in plants mainly involved in ethylene signal transduction [54]. Some studies have shown that overexpression of EIN3/EIL TFs in Arabidopsis and grapes can enhance salt resistance [17,55,56]. As an important family of nuclear TFs in the ethylene signaling pathway, *EIN3/EIL* has attracted extensive research attention. The genes in this small TF family play a crucial role in the salt tolerance and stress resistance of plants [15,17,56–59]. In *Arabidopsis*, as the core component of ethylene signal transduction, *EIN2* mutants are very sensitive to salt stress. CEND of *EIN2* attenuates the salt hypersensitivity phenotype of the mutant. The *EIN3* mutant responds normally to salt stress, while the *EIN3/EIL1* double mutant has a phenotype similar to that of the *EIN2* mutant and is extremely sensitive to high-salt conditions [60]. *EIN3* and *EIL1* regulate the ethylene response in seedlings, and seedlings have higher salt tolerance than do mature plants [15,16]. In this study, nine genes of the EIN3/EIL TF family were found to be enriched in the ethylene signaling pathway and may be related to salt tolerance in *G. maritima*. According to changes in gene expression after salt stress, three significantly up-regulated DEGs were discovered. These results extend our understanding of ethylene signaling pathway regulation in response to salt stress and help explain the function of EIN3/EIL genes in salt tolerance. However, PPI network analysis showed no interacting proteins, indicating that these three genes may be highly conserved. These results lay a

theoretical foundation for the functional verification of regulatory elements of the ethylene signaling pathway.

In this study, in addition to finding the above DEGs coding for positive regulatory TFs related to the ABA and ethylene signaling pathways, we also found DEGs coding for negative regulators in the MYC TF family related to the jasmonate signaling pathway. MYC TFs are widely expressed in animals and plants and have multiple regulatory functions [61]. When the *GhMYC4* TF in upland cotton was overexpressed in *Arabidopsis*, its salt tolerance and cold resistance were significantly improved, enhancing resistance of the plants to high-salt and drought conditions [62]. Identification and expression analysis of MYC TFs in tea plants showed that the TFs were related to amino acid anabolism under jasmonate stress and that *CsMYC1, 3, 5,* and *8* were significantly related to photosynthesis-related pathways. These results showed that under stress, jasmonate signaling may change the amino acid content and photosynthetic efficiency of tea plants through MYC family expression, thus improving adaptability under stress [63]. *MYC2*, a negative regulatory TF found in *Arabidopsis*, mediates salt sensitivity by inhibiting seed germination and delaying root growth; its knockout improves seed germination and plant root growth to better cope with salt stress [64]. These results are similar to the expression results of the two MYC family DEGs found in *G. maritima*. We found that these two DEGs interact with many proteins (Figure 9b,c), indicating that MYC TFs may play a key role in salt tolerance and stress resistance in *G. maritima*. The above results indicate that DEGs related to the MYC family in *G. maritima* participate most in protein regulation and mediation, which lays a theoretical foundation for explaining the function of salt resistance genes in *G. maritima*. These results have improved the theoretical basis for our subsequent functional verification experiments.

## 5. Conclusions

In this study, 1466 TFs genes were counted by analyzing the TF families in the TF data of the leaves of *G. maritima* under different salt concentrations; these genes were distributed across 57 TF families. After trend analysis, a total of 820 DEGs were counted, mainly distributed across 8 expression modules. After qRT-PCR expression verification, it was found that the TFs *ABF2* (*Unigene0078257*) were related to ABA, *EIN3* (*Unigene0000457* and *Unigene0012139*) and *EIL1* (*Unigene0042139*) were related to ethylene regulation, and *MYC1/2* (*Unigene0009899* and *Unigene0027167*) were related to the bHLH TF family. These TF genes may play an important role in regulating salt tolerance in *G. maritima*. These results provided a theoretical basis for improving the salt tolerance and stress resistance of plants and exploring the genetic resources for salt tolerance in halophytes.

**Supplementary Materials:** The following supporting information can be downloaded at: https://www.mdpi.com/article/10.3390/agriculture13071404/s1, Figure S1: Physiological and biochemical indexes of G. maritima leaves under salt stress.; Figure S2: Agilent 2100 RNA 6000 Nano kit detection Table S1: the quality of the isolated RNA samples; Table S2: Characteristics of G. maritima transcriptome sequencing; Table S3: Sequencing data of G. maritima leaves under different salt concentrations; Table S4: Primer sequences used for real-time fluorescence quantitative analysis.

**Author Contributions:** F.S. designed the experiment, R.G., Z.W. and M.Y. performed the experiments, R.G. and F.T. analyzed the data, R.G. wrote the manuscript. All authors have read and agreed to the published version of the manuscript.

**Funding:** This research was funded by the Science and Technology Project of the Inner Mongolia Autonomous Region (2020GG0063) and the Key Projects in Science and Technology of Inner Mongolia (2021ZD0031).

**Data Availability Statement:** Datasets are available at the NCBI project PRJNA845001. The transcriptome datasets are available in the NCBI Short Read Archive (SRA) database with the accession numbers SAMN28853762, SAMN28853763, SAMN28853764, SAMN28853765, SAMN28853766, SAMN28853767, SAMN28853768, SAMN28853769, and SAMN28853770.

**Acknowledgments:** Thanks to the Key Laboratory of Grassland Resources, Ministry of Education of P.R. of China for providing facilities.

**Conflicts of Interest:** The authors declare no conflict of interest.

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
