# Peer review of "Leaf Transcription Factor Family Analysis of Halophyte Glaux maritima under Salt Stress"

_agriculture, doi:10.3390/agriculture13071404_

Round 1

Reviewer 1 Report

For Authors:

The manuscript entitled “Leaf Transcription Factor Family analysis of Halophyte Glaux maritima under Salt Stress” by Gu Rui J. et al. has been reviewed. MS provides basic data about transcriptome analysis of G. maritima leaves under different salt concentrations, and it could be useful for improving the salt tolerance and stress resistance of plants.

Minor comments to Authors:

          Figure 2, Figure 3, Figure 4, Figure 5, .... should be improved with better resolution, and the size of the words should be visible. Please correct that.

          Line 88-89: “The collection of plant seedlings and all methods were performed in accordance with the relevant guidelines and regulations” – Please add references for guidelines and regulations which are used in the MS.

          Generally MM section must be improved with detailed information. For example:

Section 2.1 don’t have information for RNA isolation, what is the mass of the homogenized tissue and other experimental conditions which are applied.

Author Response

Thanks! the honored editor and the reviewers! We really appreciate your comments and suggestions on revising the manuscript.
We have studied reviewer’s comments carefully and have explained to clarify many related issues and made an appropriate revision in the manuscript. We think that the response has given the clear explanations on the confusing issues and the revised manuscript has been improved as a result of the constructive advices.

1.Figure 2, Figure 3, Figure 4, Figure 5, .... should be improved with better resolution, and the size of the words should be visible. Please correct that.

Response: Thank you for your important comments! We had made modifications to each of the image questions cited in the article and added annotations to improve self-explanatory. Revised as follows:

Figure 2. KEGG pathway enrichment analysis of the transcription factor genes. Note: under salt stress, a total of 95 TF genes were successfully annotated in the KEGG database for Glaux maritima transcription factor genes. Among the three KEGG pathways, Plant hormone signaling pathway (66 genes), MAPK signaling pathway -plant (34 genes) and plant pathogen interaction (18 genes), the number of genes annotated is the largest.

Figure 3. GO enrichment analysis of the transcription factor genes. Note: under salt stress, A total of 54 functions were annotated, and 1191 genes were enriched in GO annotation and enrichment analysis. Among the three ontological processes, bindng (GO:0005488) and nuclear acid binding transcription factor activity (GO:0001071) had the highest distribution of Molecular Function; (GO:0005623), cell part (GO:0044464), and organelle (GO:0043226) had the highest distribution of Cell Component; cellular process (GO:0009987), metabolic process (GO:008152), regulation of biological process (GO:0050789), and biological regulation (GO:0065007) had the highest distribution of Biological Process.

Figure 4. Gene expression module diagram for trend analysis of all transcription factors. Note: After trend analysis, there were a total of 820 differentially expressed genes distributed in 8 expression modules. They were profile0(83 genes), profile1(239 genes), profile2(60 genes), profile3(62 genes), profile4(28 genes), profile5(62 genes), profile6(222 genes), and profile7(64 genes). In addition, the gene expression in the four modules of profile 0, 1, 6, and 7 showed significant differences after different concentrations of salt stress (p < 0.05). N0:0 mM/L NaCl stress; N3:600mM/L NaCl stress; N4:800mM/L NaCl stress.

Figure 5. Differentially expressed transcription factor gene statistics. Note: The figure shows the distribution of 820 transcription factor genes in the top 11 TF families in the Glaux maritima. The 11 TF families were ERF (96, 11.71%), bHLH (75, 9.15%), WRKY (67, 8.17%), NAC (65, 7.93%), MYB-related (49, 5.98%), C2H2 (47, 5.73%), MYB (45, 5.49%), bZIP (37, 4.51%), C3H (25, 3.05%), GRAS (25, 3.05%), and HD-ZIP (22, 2.68%) families.

Figure 6. Numbers of genes in the top 10 transcription factor families. Note: Statistics on the total number of transcription factors in the top 10 transcription factor families of Glaux maritima and the number of differentially expressed genes in trend analysis. The 10 transcription factor families were ERF (TF:133 genes, DEGs:96 genes),bHLH(TF:123 genes, DEGs:75 genes),MYB-related(TF:95 genes, DEGs:49genes),NAC(TF:94 genes ,DEGs: 65 genes),WRKY(TF:84genes DEGs:,67 genes),C2H2(TF:81 genes, DEGs: 47 genes),bZIP(TF:70genes ,DEGs:37genes),MYB(TF:66 genes, DEGs:45 genes),C3H(TF:53 genes, DEGs:25 genes)and GRAS(TF:48 genes, DEGs:25 genes) .TF: transcription factor, DEGs: differentially expressed genes.

Figure 7. Heat map of differential transcription factor gene expression and log2 FPKM expression trends. Note: a: Heat map of 33 up-regulated differentially expressed genes under different salt concentrations; b: log2 FPKM expression trends of 33 upregulated differentially expressed genes under different salt concentrations; c: Heat map of 35 down-regulated differentially expressed genes under different salt concentrations; d: log2 FPKM expression trends of 35 down-regulated differentially expressed genes under different salt concentrations. N0:0 mM/L NaCl stress; N3:600mM/L NaCl stress; N4:800mM/L NaCl stress.

Figure 9. PPI network analysis on DEGs found in ABFs and MYC transcription factor families. Note: a: The ABFs transcription factor related differential gene (Unigene0078257) interacted with 106 genes, forming a total of 472 interactions; b: The MYC transcription factor related differential gene (Unigene0009899) interacted with 349 genes, forming a total of 4531 interactions; c: The MYC transcription factor related differential gene (Unigene0027167) interacted with 348 genes, forming a total of 3526 interactions.

Figure 10. Differentially expressed genes involved in plant hormone signal transduction. Note: N0:0 mM/L NaCl stress; N3:600mM/L NaCl stress; N4:800mM/L NaCl stress.

Figure 11. Differentially expressed genes enriched in the MAPK signaling pathway - plant. Note: N0:0 mM/L NaCl stress; N3:600mM/L NaCl stress; N4:800mM/L NaCl stress.

2.Line 88-89: “The collection of plant seedlings and all methods were performed in accordance with the relevant guidelines and regulations” – Please add references for guidelines and regulations which are used in the MS.

Response: Thank you for your important comments! We had removed this sentence from the text. Due to the previous submission process where the assistant editor requested an explanation of the reasonableness of the sampling of the plants, this sentence needs to be added. This magazine has not made any requirements in this regard, so it has been removed.

3.Generally MM section must be improved with detailed information. For example:

Section 2.1 don’t have information for RNA isolation, what is the mass of the homogenized tissue and other experimental conditions which are applied.

Response: Thank you for your important comments! The RNA extraction method was based on the steps in the reagent kit, so there was no description. The reference literature we cited provides a detailed introduction to the RNA extraction steps. At the same time, we had added information on the quality of extracted RNA in the attachment file. Here is the information on RNA quality.

We also added corresponding references to RNA extraction. Modify as follows: Total RNA was extracted using a TRIzol reagent kit (Invitrogen, Carlsbad, CA, USA), as per the manufacturer's instructions (Pandey, et al,2022). To verify the integrity of the RNA, we processed it through RNase-free agarose gel electrophoresis and then analyzed the results using an Agilent 2100 Bioanalyzer (Agilent Technologies, Palo Alto, CA, USA). Following total RNA isolation, DynabeadsTM Oligo(dT)25 beads (Thermo Fisher Scientific, Waltham, MA, USA) were used to remove rRNA from eukaryotic mRNA, and a Ribo-ZeroTM Magnetic Kit was used to do the same for bacterial mRNA (Illumina, San Diego, CA, USA) (Attached Figure2, Supplementary Table1).

Supplementary Table 1. the quality of the isolated RNA samples

number

Sample name

Concentration (ng/ul)

Volume (ul)

Total(ug)

1

N0-1

258

33

8.51

2

N0-2

255

33

8.42

3

N0-3

296

33

9.77

4

N3-1

357

33

11.78

5

N3-2

303

33

10

6

N3-3

284

33

9.37

7

N4-1

228

33

7.52

8

N4-2

387

33

12.77

9

N4-3

173

33

5.71

Note: N0:0mM/L NaCl stress; N3:600mM/L NaCl stress; N4:800mM/L NaCl stress.

Attached Figure2 Agilent 2100 RNA 6000 Nano kit detection

Note:1-3: N0:0mM/L NaCl stress;4-6: 600mM/L NaCl stress;7-9: 800mM/L NaCl stress;

Reviewer 2 Report

The current manuscript has a significant contribution to understanding the genetic factors regulating salt tolerance in halophytes. I strongly believe that these candidate genetic factors (genes) can be used in molecular breeding programs such as genetic engineering or CRISPR/Cas9. I have a few minor comments. 

1. The concentration and duration of salt stress should be added in the abstract section.

2. Please write the full form of each term and then use abbreviations. As an example ' WRKY, ERF, NAC, GO, MAPK, KEGG'. 

3. The authors did not explain the function of the candidates' genes identified in the study. Should be added in the abstract section. 

4. What is the novelty of this study? How this study is different from other studies? Please explain this point at the end of the introduction section.

5. Lines, 78-89 need more detail. Meanwhile, the authors exposed plants to salt stress only for 24 hours. Explain the reason for short-term exposure to salt stress.

6. Improve the quality of Figures 2 and 4. 

7. Add more relevant references in the discussion section. 

8. Manuscript should be proofread to correct typos and grammatical err

The manuscript should be proofread. Minor English editing is required. 

Author Response

Thanks! the honored editor and the reviewers! We really appreciate your comments and suggestions on revising the manuscript.
We have studied reviewer’s comments carefully and have explained to clarify many related issues and made an appropriate revision in the manuscript. We think that the revised manuscript has been improved as a result of the constructive advices.
1. The concentration and duration of salt stress should be added in the abstract section.

Response: Thank you for your important comments! We had added the research purpose and novelty of this article in the abstract, and the revised abstract is as follows: In this study, 1466 transcription factors were identified by transcriptome sequencing analysis of Glaux maritima leaves after salt stress (0, 600 and 800mM/L NaCl).

  1. Please write the full form of each term and then use abbreviations. As an example ' WRKY, ERF, NAC, GO, MAPK, KEGG'.

Response: Thank you for your important comments! We had completed the full name of the abbreviation mentioned for the first time in the manuscript. Modify as follows: NAC (NAM,ATAF1/2,CUC2), ERF/AP2(Ethylene Response factor /APETALA2), bZIP (basic leucine-zipper), bHLH(Basic Helix-Loop-Helix), MYB(v-myb avian mye1oblastosis viral oncogene homolog), MAPK (Mitogen-activated protein kinases) signaling pathway, Gene Ontology (GO) and Kyoto Encyclopedia of Genes and Genomes (KEGG).

  1. The authors did not explain the function of the candidates' genes identified in the study. Should be added in the abstract section.

Response: Thank you for your important comments! We had added the research purpose and novelty of this article in the abstract, and the revised abstract is as follows: KEGG analysis revealed four genes with significant positive regulation, ABF2 (Unigene0078257) in the ABA signaling pathway, EIN3 (Unigene0000457 and Unigene0012139), and EIL1 (Unigene0042139) involved in ethylene signal transduction, and two with negative regulation, MYC1/2 (Unigene0009899 and Unigene0027167) in the main regulator of Jasmonic acid signal transduction.

  1. What is the novelty of this study? How this study is different from other studies? Please explain this point at the end of the introduction section.

Response: Thank you for your important comments! We had improved the last paragraph of the introduction and proposed the novelty of this article. The content of the article is as follows: “As an important agricultural and animal husbandry area in northern China, Inner Mongolia has been experiencing an annual increase in saline alkali land area due to its arid climate and the fact that irrigation is the main planting method. Therefore, there is an urgent need to study salt tolerant mesophytes that are suitable for growing in the ecological environment of the region, in order to provide genetic resources for culti-vating salt tolerant and highly resistant agricultural and pastoral crops. Based on the previous analysis of the morphology, physiology and Transcriptome of Glaux maritima, this study analyzed the salt-tolerance-related TFs of the salt-secreting halophyte Glaux maritima at the transcription level and identified six important genes distributed in the EIN, bHLH, and bZIP TF families. Clustering and expression analyses of these genes revealed that they play key roles in the response to high-salt conditions and could pro-vide valuable genetic resources for cultivating salt-tolerant crops.”

  1. Lines, 78-89 need more detail. Meanwhile, the authors exposed plants to salt stress only for 24 hours. Explain the reason for short-term exposure to salt stress.

Response: Thank you! We have rewritten the paragraph mentioned. Modify as follows: “Inner Mongolia Agricultural University's Key Laboratory of Grassland and Resources of the Ministry of Education cultivated wild G. maritima seedlings in 1/2 Hoagland solution in a greenhouse for 15 days (wild G. maritima seedlings were collected from the Hailiutu base of Inner Mongolia Agricultural University Science and Technology Park, Bikeqi Town, Tumet Left Banner, Hohhot, Inner Mongolia, N:40 ° 38 ′, E:111 ° 28 ′, Height:1060 m.). The optimal growing conditions included a steady 25 °C temperature, 60% relative humidity, and 1600 lx of light. Selected G. maritima seedlings were subjected to NaCl concentrations of 0(N0), 600(N3), and 800(N4) mM for 24 hours. Moreover, leaf-samples of G. maritima were collected in triplicates for molecular analysis and directly frozen in liquid nitrogen and kept at -80℃ for RNA isolation (Pandey, et al,2022).”

The design of this study mainly refers to the following references: Zhang et al (https://doi.org/10.1016/j.ecoenv.2020.110259),2020;Ben et al (https://doi.org/10.1007/s10725-022-00805-0),2022;Li,2022(Comparative Analysis of Transcriptome of Two Glycyrrhiza Species in Response to Salt Stress and Establishment of their Shoot Apex-Based Genetic Transformation System.(dissertation)Lanzhou University, Gansu. China) and Chen,2021(Functional Analysis of Alfalfa MsABF2 in abiotic stress tolerance. (dissertation) Harbin Normal University, Heilongjiang, China). Salt stress is applied to Glaux maritima seedlings based on the stress time and results used in the literature.

  1. Improve the quality of Figures 2 and 4.

Response: Thank you for your important comments! We had made modifications to each of the image and table questions cited in the article and added annotations to improve self-explanatory. Revised as follows:

Figure 2. KEGG pathway enrichment analysis of the transcription factor genes. Note: under salt stress, a total of 95 TF genes were successfully annotated in the KEGG database for Glaux maritima transcription factor genes. Among the three KEGG pathways, Plant hormone signaling pathway (66), MAPK signaling pathway -plant (34) and plant pathogen interaction (18), the number of genes annotated is the largest.

Figure 4. Gene expression module diagram for trend analysis of all transcription factors. Note: After trend analysis, there were a total of 820 differentially expressed genes distributed in 8 expression modules. They were profile0(83 genes), profile1(239 genes), profile2(60 genes), pro-file3(62 genes), profile4(28 genes), profile5(62 genes), profile6(222 genes), and profile7(64 genes). In addition, the gene expression in the four modules of profile 0, 1, 6, and 7 showed significant differences after different concentrations of salt stress (p < 0.05). N0:0 mM/L NaCl stress; N3:600mM/L NaCl stress; N4:800mM/L NaCl stress.

  1. Add more relevant references in the discussion section.

Response: Thank you for your important comments! We had supplemented a large amount of literature during the discussion.

  1. Manuscript should be proofread to correct typos and grammatical err

Response: Thank you for your important comments! We had improved the grammar errors in the manuscript.

9.Comments on the Quality of English Language

The manuscript should be proofread. Minor English editing is required.

Response: Thank you for your important comments! We had found Editage (www.editage.cn) to polish the manuscript for English language editing.

Reviewer 3 Report

To,

The Editor-in-chief,

Agriculture, MDPI,

Manuscript ID: agriculture-2442618

Subject: Submission of comments of the manuscript in “Agriculture"

Dear Editor-in-chief Agriculture, MDPI,

Thank you very much for the invitation to consider a potential reviewer for the manuscript (ID: agriculture-2442618). My comments responses are furnished below as per each reviewer’s comments.

In the reviewed manuscript, 1466 transcription factors were identified by transcriptome sequencing analysis of the leaves of the halophyte Glaux maritima after salt stress. Their genes were mainly distributed in 57 transcription factor families, including ERF, bHLH, MYB-related, NAC, and WRKY. KEGG and GO analyses showed significant enrichment in 14 pathways, including plant hormone signal transduction and the MAPK signaling pathway, with a total of 54 functions annotated. Gene expression analysis showed 820 differentially expressed genes mainly distributed in 11 transcription factor families, including ERF, bHLH, WRKY, and NAC, and 8 expression modules. KEGG analysis revealed four genes with significant positive regulation, ABF2 (Unigene0078257), EIN3 (Unigene0000457 and Unigene0012139), and EIL1 (Unigene0042139), and two with negative regulation, MYC1/2 (Unigene0009899 and Unigene0027167). Protein–protein interaction networks suggested ABF2 and MYC1/2 as important transcription factors regulating G. maritima salt tolerance. This study provides a theoretical basis for mining salt tolerance genetic resources from halophytes and lays a foundation for im-proving the salt tolerance of plants. Therefore, it might be conditionally accepted subject to major revision. Authors have to improve their manuscripts with many non-clear meanings, inaccuracies and inconsistencies, and the authors need to address the following issues.

  1. I have read the entire manuscript and my initial comment is that manuscript is poorly written. I have significant concerns about the grammar and vocabulary of the manuscript; therefore, I recommend the authors to used an English proofreading service.
  2. The writing style of the paper is very poor. There are lots of grammatical mistakes. Long sentences with noticeable grammatical mistakes are frequently present throughout the manuscript.
  3. In abstract, the author should add more scientific findings.
  4. Introduction part is not impressive and systematic. In the introduction part, the authors should elaborate the scientific issues in the plants research.
  5. Figure quality is not good and text is not readable for instance, Figure 3, 4, 7, 8, 9, 10 and 11.
  6. Results section has again serious flaws in the presentation.
  7. The discussion should be interpreted with the results as well as discussed in relation to the present literature. Comparison of the present results with other similar findings in the literature should be discussed in more detail. This is necessary in order to place this work together with other work in the field and to give more credibility to the present results.
  8. The conclusion section is very lengthy. The author should emphasize this in a better way.
  9. References: shall have to correct the whole References according to the ”Instructions for the Authors”, e.g. the Journal name is in italics, the year must be bold and you shall have to use the abbreviated name of the Journals cited.

To,

The Editor-in-chief,

Agriculture, MDPI,

Manuscript ID: agriculture-2442618

Subject: Submission of comments of the manuscript in “Agriculture"

Dear Editor-in-chief Agriculture, MDPI,

Thank you very much for the invitation to consider a potential reviewer for the manuscript (ID: agriculture-2442618). My comments responses are furnished below as per each reviewer’s comments.

In the reviewed manuscript, 1466 transcription factors were identified by transcriptome sequencing analysis of the leaves of the halophyte Glaux maritima after salt stress. Their genes were mainly distributed in 57 transcription factor families, including ERF, bHLH, MYB-related, NAC, and WRKY. KEGG and GO analyses showed significant enrichment in 14 pathways, including plant hormone signal transduction and the MAPK signaling pathway, with a total of 54 functions annotated. Gene expression analysis showed 820 differentially expressed genes mainly distributed in 11 transcription factor families, including ERF, bHLH, WRKY, and NAC, and 8 expression modules. KEGG analysis revealed four genes with significant positive regulation, ABF2 (Unigene0078257), EIN3 (Unigene0000457 and Unigene0012139), and EIL1 (Unigene0042139), and two with negative regulation, MYC1/2 (Unigene0009899 and Unigene0027167). Protein–protein interaction networks suggested ABF2 and MYC1/2 as important transcription factors regulating G. maritima salt tolerance. This study provides a theoretical basis for mining salt tolerance genetic resources from halophytes and lays a foundation for im-proving the salt tolerance of plants. Therefore, it might be conditionally accepted subject to major revision. Authors have to improve their manuscripts with many non-clear meanings, inaccuracies and inconsistencies, and the authors need to address the following issues.

  1. I have read the entire manuscript and my initial comment is that manuscript is poorly written. I have significant concerns about the grammar and vocabulary of the manuscript; therefore, I recommend the authors to used an English proofreading service.
  2. The writing style of the paper is very poor. There are lots of grammatical mistakes. Long sentences with noticeable grammatical mistakes are frequently present throughout the manuscript.
  3. In abstract, the author should add more scientific findings.
  4. Introduction part is not impressive and systematic. In the introduction part, the authors should elaborate the scientific issues in the plants research.
  5. Figure quality is not good and text is not readable for instance, Figure 3, 4, 7, 8, 9, 10 and 11.
  6. Results section has again serious flaws in the presentation.
  7. The discussion should be interpreted with the results as well as discussed in relation to the present literature. Comparison of the present results with other similar findings in the literature should be discussed in more detail. This is necessary in order to place this work together with other work in the field and to give more credibility to the present results.
  8. The conclusion section is very lengthy. The author should emphasize this in a better way.
  9. References: shall have to correct the whole References according to the ”Instructions for the Authors”, e.g. the Journal name is in italics, the year must be bold and you shall have to use the abbreviated name of the Journals cited.

Author Response

Thanks! the honored editor and the reviewers! We really appreciate your comments and suggestions on revising the manuscript.
We have studied reviewer’s comments carefully and have explained to clarify many related issues and made an appropriate revision in the manuscript. We think that the revised manuscript has been improved as a result of the constructive advices.
1. I have read the entire manuscript and my initial comment is that manuscript is poorly written. I have significant concerns about the grammar and vocabulary of the manuscript; therefore, I recommend the authors to used an English proofreading service.

2.The writing style of the paper is very poor. There are lots of grammatical mistakes. Long sentences with noticeable grammatical mistakes are frequently present throughout the manuscript.

Response: Thank you for your important comments! We have carefully read and revised the shortcomings in grammar and expression mentioned by the reviewers in the manuscript, and we had found Editage (www.editage.cn) to polish the manuscript for English language editing.

3.In abstract, the author should add more scientific findings.

Response: Thank you for your important comments! We had added the research purpose and novelty of this article in the first to third sentences of the abstract and the revised abstract is as follows: “The reduction of crop yield caused by soil salinization has become a global problem. Halophytes improve saline alkali soil and the halophyte transcription factors that regulate salt stress and are crucial for improving salt tolerance. In this study, 1466 transcription factors were identified by transcriptome sequencing analysis of Glaux maritima leaves after salt stress (0, 600 and 800mM/L NaCl). Their genes were distributed across 57 transcription factor families. KEGG and GO analyses showed significant enrichment in 14 pathways with a total of 54 functions annotated. Gene expression analysis showed 820 differentially expressed genes distributed in 11 transcription factor families, including ERF, bHLH, WRKY, and NAC, and 8 expression modules. KEGG analysis revealed four genes with significant positive regulation, ABF2 (Unigene0078257) in the ABA signaling pathway, EIN3 (Unigene0000457 and Unigene0012139), and EIL1 (Unigene0042139) involved in ethylene signal transduction, and two with negative regulation, MYC1/2 (Unigene0009899 and Unigene0027167) in the main regulator of Jasmonic acid signal transduction. Protein–protein interaction networks suggested ABF2 and MYC1/2 as important transcription factors regulating G. maritima salt tolerance. Overall, the salt tolerant transcription factors discovered in this study provide genetic resources for plant salt tolerance inheritance, and lay a theoretical foundation for the study of the salt tolerant molecular mechanism of the halophyte Glaux maritima.”

4.Introduction part is not impressive and systematic. In the introduction part, the authors should elaborate the scientific issues in the plants research.

Response: Thank you for your important comments! We had improved the last paragraph of the introduction and proposed the novelty of this article. The content of the article is as follows: “As an important agricultural and animal husbandry area in northern China, Inner Mongolia has been experiencing an annual increase in saline alkali land area due to its arid climate and the fact that irrigation is the main planting method. Therefore, there is an urgent need to study salt tolerant mesophytes that are suitable for growing in the ecological environment of the region, in order to provide genetic resources for cultivating salt tolerant and highly resistant agricultural and pastoral crops. Based on the previous analysis of the morphology, physiology and Transcriptome of Glaux maritima, this study analyzed the salt-tolerance-related TFs of the salt-secreting halophyte Glaux maritima at the transcription level and identified six important genes distributed in the EIN, bHLH, and bZIP TF families. Clustering and expression analyses of these genes revealed that they play key roles in the response to high-salt conditions and could provide valuable genetic resources for cultivating salt-tolerant crops.”

5.Figure quality is not good and text is not readable for instance, Figure 3, 4, 7, 8, 9, 10 and 11.

Response: Thank you for your important comments! We had made modifications to each of the image and table questions cited in the article and added annotations to improve self-explanatory. Revised as follows:

Figure 3. GO enrichment analysis of the transcription factor genes. Note: under salt stress, A total of 54 functions were annotated, and 1191 genes were enriched in GO annotation and enrichment analysis. Among the three ontological processes, bindng (GO:0005488) and nuclear acid binding transcription factor activity (GO:0001071) had the highest distribution of Molecular Function; (GO:0005623), cell part (GO:0044464), and organelle (GO:0043226) had the highest distribution of Cell Component; cellular process (GO:0009987), metabolic process (GO:008152), regulation of biological process (GO:0050789), and biological regulation (GO:0065007) had the highest distribution of Biological Process.

Figure 4. Gene expression module diagram for trend analysis of all transcription factors. Note: After trend analysis, there were a total of 820 differentially expressed genes distributed in 8 expression modules. They were profile0(83 genes), profile1(239 genes), profile2(60 genes), pro-file3(62 genes), profile4(28 genes), profile5(62 genes), profile6(222 genes), and profile7(64 genes). In addition, the gene expression in the four modules of profile 0, 1, 6, and 7 showed significant differences after different concentrations of salt stress (p < 0.05). N0:0 mM/L NaCl stress; N3:600mM/L NaCl stress; N4:800mM/L NaCl stress.

Figure 7. Heat map of differential transcription factor gene expression and log2 FPKM expression trends. Note: a: Heat map of 33 up-regulated differentially expressed genes under different salt concentrations; b: log2 FPKM expression trends of 33 upregulated differentially expressed genes under different salt concentrations; c: Heat map of 35 down-regulated differentially expressed genes under different salt concentrations; d: log2 FPKM expression trends of 35 down-regulated differentially expressed genes under different salt concentrations. N0:0 mM/L NaCl stress; N3:600mM/L NaCl stress; N4:800mM/L NaCl stress.

Figure 8. Phylogenetic tree analysis of genes in the ABF, MYC, and EIN3/EIL transcription factor families. Note: The genes were identified via trend analysis of transcription factors in Glaux maritima and are related to plant hormone signal transduction. A: Phylogenetic tree analysis of ABF2 transcription factors. (Unigene0003302, Unigene0007479, Unigene0007480, Unigene0007482, and Unigene0078257);b: Phylogenetic tree analysis of MYC1 and MYC2 transcription factors (Unigene0006978, Unigene0009899, Unigene0027154, Unigene0027167, Unigene0046064, Unigene0069618, and Unigene0077023);c: Phylogenetic tree analysis of EIN3 tran-scription factors(Unigene0000457, Unigene0012139, Unigene0036810,Unigene0036956, Unigene0036957, Unigene0042139, Unigene0071273, Unigene0071274, and Unigene0071276).

Figure 9. PPI network analysis on DEGs found in ABFs and MYC transcription factor families. Note: a: The ABFs transcription factor related differential gene (Unigene0078257) interacted with 106 genes, forming a total of 472 interactions; b: The MYC transcription factor related differential gene (Unigene0009899) interacted with 349 genes, forming a total of 4531 interactions; c: The MYC transcription factor related differential gene (Unigene0027167) interacted with 348 genes, forming a total of 3526 interactions.

Figure 10. Differentially expressed genes involved in plant hormone signal transduction. Note: N0:0 mM/L NaCl stress; N3:600mM/L NaCl stress; N4:800mM/L NaCl stress.

Figure 11. Differentially expressed genes enriched in the MAPK signaling pathway - plant. Note: N0:0 mM/L NaCl stress; N3:600mM/L NaCl stress; N4:800mM/L NaCl stress.

6.Results section has again serious flaws in the presentation.

Response: Thank you for your important comments! We had carefully read and revised the expression of the results, such as: “

3.2. KEGG and GO enrichment analysis of all transcription factors

In order to better understand the main biochemical and signal transduction pathways related to the obtained transcription factors, KEGG pathway analysis was conducted on transcription factor family (Figure 2). A total of 95 TF genes were significantly enriched in 14 pathways, including plant hormone signal transduction (66 unigenes), MAPK (Mitogen-activated protein kinases) signaling pathway (34 unigene), plant–pathogen interaction (18 unigene), circadian rhythm (5 unigene), base exception repair (1 unigene), and DNA replication (1 unigene). These annotations provide im-portant information for studying the specific functions, processes, and mechanisms involved in the salt tolerance of halophytes.”

7.The discussion should be interpreted with the results as well as discussed in relation to the present literature. Comparison of the present results with other similar findings in the literature should be discussed in more detail. This is necessary in order to place this work together with other work in the field and to give more credibility to the present results.

Response: Thank you for your important comments! We had carefully read and revised the discussion in response to the suggestions put forward by the review experts regarding the description in the discussion.

For example, we will modify the fourth paragraph of the discussion section follows: as: “Transcriptional regulation plays an important role in plant resistance to stress. Most regulation will be realized through interactions between specific TFs and corresponding cis-acting elements. TFs, as DNA-binding proteins that specifically interact with cis-acting elements in the gene promoter region, will activate or inhibit transcription on their own or with other proteins so that genes are expressed in specific places at specific times. Many different TF families related to salt stress, including WRKY, avian myeloblastosis viral oncogene homolog (MYB), NAM, ATAF1/2 and CUC2 (NAC), basic leucine zipper (bZIP), APETALA2/ethylene response factor (AP2/ERF), and basic helix loop helix (bHLH), among others, have been identified by analyzing the whole genomes of different plants [39]. With the increasing global saline-alkali land area, halophytes are receiving increasing attention from researchers. Previous studies on the salt tolerance characteristics and molecular mechanisms of halophytes have been conducted through phenotypic, physiological, and molecular levels [22-26]. Halophytes could provide better genetic resources for improving plant salt tolerance. At present, the salt-tolerant TF families found in halophytes mainly include AP2/EREBP, MYB, WRKY, NAC, HD ZIP, zinc finger domain (ZnF), G-box binding proteins, 14-3-3 protein analogs, MADS box proteins, and bHLH. These TFs show different expression patterns under salt stress and have specific or universal response mechanisms for certain times and locations. Additionally, they participate in complex regulatory networks in plants [40–45]. This is similar to the results of the study on the salt tolerant transcription factor family in the halophytic plant G. maritima. This study also found that transcription factors in the ERF, bHLH, and bZIP transcription factor families play a crucial role in the adaptation of G. maritima to high salt environments. In this study, by combining KEGG and GO enrichment analysis and trend analysis, we analyzed the DEGs in these transcription factor families, and found that 6 salt tolerant key genes in the EIN, bZIP and bHLH transcription factor families were expressed in the Plant hormone signal pathway. These was similar to the results of regulatory genes in the transcription factor families such as ERF, MYB, bZIP, and bHLH in model plants such as Arabidopsis, tobacco, soybeans, Medicago truncatula and Zea mays, which enhance plant salt tolerance and affect plant salt tolerance [46-49]. The molecular mechanism of salt tolerance in the halophyte G. maritima may be directly related to the regulation of these transcription factors.”

8.The conclusion section is very lengthy. The author should emphasize this in a better way.

Response: Thank you for your important comments! We had rephrased the conclusion as: “In this study, 1466 transcription factors genes were counted by analyzing the transcription factor families in the Transcriptome data of the leaves of G. maritima under different salt concentrations; these genes were distributed across 57 transcription factor families. After trend analysis, a total of 820 DEGs were counted, mainly distributed across 8 expression modules. After qRT-PCR expression verification, it was found that the transcription factors ABF2 (Unigene0078257) was related to ABA, EIN3 (Unigene0000457 and Unigene0012139) and EIL1 (Unigene0042139) were related to ethylene regulation, and MYC1/2 (Unigene0009899 and Unigene0027167) was related to the bHLH transcription factor family. These may be important transcription factors regulating salt tolerance in G. maritima. These results from this study provide a theoretical basis for mining salt-tolerant genetic resources from halophytes and lay a foundation for improving the salt tolerance and stress resistance of plants.”

9.References: shall have to correct the whole References according to the Instructions for the Authors”, e.g. the Journal name is in italics, the year must be bold and you shall have to use the abbreviated name of the Journals cited.

Response: Thank you for your important comments! We had checked the format of all references and standardized all formats. The format is modified as: “Lindemose, S.; O’Shea, C.; Jensen, M. K.; Skriver, K. Structure, function and networks of transcription factors involved in abiotic stress responses. Int J Mol Sci .2013,14:5842–5878. DOI:10.3390/ijms14035842

Reviewer 4 Report

Abstract – well written

Additional space in Keywords section

M&M

The first sentence of this section should be rewritten. It s not clear.

Lack of “)” after N and E number.(line 83)

“Take plant 85 leaf samples, put them into liquid nitrogen for quick freezing, and store them at - 80 ℃ 86 for RNA extraction and transcriptome sequencing”.- is this instructon or method description? This paragraph need rewriting

“The collection of plant seedlings and all methods were performed in accordance with 88 the relevant guidelines and regulations.” This sentence should be rewritten

qRT-PCR should meet the MIQE criteria descripted by Bustin et al., 2009. Only one references gene?

67 line (Yuan et al) : I think the references here should be shown in different way: keep the consequences in ref style

Please provide clear goal of your study at the end of introduction section.

Results

Figure 1. Please provide more details in the description of this figure

Figure 2. Low quality. Need improvement.

Figure 3 . Low quality. Need improvement.

Figure 4 . Low quality. Need improvement.

Captions for all figure should be better explain and descript

Figure 7 . Low quality. Need improvement.

Lines 242-245 – this text should be transfer to the Discussion section

Lines 258-260 – this text should be transfer to the Discussion section

Lines 275-278 – this text should be transfer to the Discussion section

Reference section – a mess

Author Response

Thanks! the honored editor and the reviewers! We really appreciate your comments and suggestions on revising the manuscript.
We have studied reviewer’s comments carefully and have explained to clarify many related issues and made an appropriate revision in the manuscript. We think that the response has given the clear explanations on the confusing issues and the revised manuscript has been improved as a result of the constructive advices.

Abstract – well written

Additional space in Keywords section

M&M

1.The first sentence of this section should be rewritten. It s not clear.

Response: Thank you for your important comments! We had rephrased The first sentence of this section as: “Wild G. maritima seedlings were cultivated with 1/2 Hoagland nutrient solution for 15 days in the Key Laboratory of the Ministry of Education, College of Grassland, Resources and Environment, Inner Mongolia Agricultural University (wild G. maritima seedlings were collected from the Hailiutu base of Inner Mongolia Agricultural University Science and Technology Park, Bikeqi Town, Tumet Left Banner, Hohhot, Inner Mongolia, N:40 ° 38 ′, E:111 ° 28 ′, Height:1060 m.). ”

2.Lack of “)” after N and E number.  (line 83)

Response: Thanks! We had added the missing “)” after N and E number. (line 96).

3.“Take plant 85 leaf samples, put them into liquid nitrogen for quick freezing, and store them at - 80 ℃ 86 for RNA extraction and transcriptome sequencing”.- is this instructon or method description? This paragraph need rewriting

Response: Thank you for your important comments! We had rephrased This sentence as: “Moreover, leaf-samples of G. maritima were collected in triplicates for molecular analysis and directly frozen in liquid nitrogen and kept at -80℃ for RNA isolation.”

4.“The collection of plant seedlings and all methods were performed in accordance with 88 the relevant guidelines and regulations.” This sentence should be rewritten

Response: Thank you for your important comments! We had removed this sentence from the text. Due to the previous submission process where the assistant editor requested an explanation of the reasonableness of the sampling of the plants, this sentence needs to be added. This magazine has not made any requirements in this regard, so it has been removed.

5.qRT-PCR should meet the MIQE criteria descripted by Bustin et al., 2009. Only one references gene?

Response: Thank you for your important comments! In this study, reverse transcription experiments were strictly carried out according to the method mentioned by MIQE. The method used the mentioned primer as the internal reference primer to verify the gene, as mentioned in the attached information table. The expression measurement of the target gene in each sample was repeated 3 times. Based on the research results, this study conducted a pathway gene heatmap.

The primers in the attachment are as follows:

Supplementary Table4. Primer sequences used for real-time fluorescence quantitative analysis

Gene

Primer sequence (5'to3')

Fragment size(bp)

β-Actin -F

TGGCACTTGATTACGAGCAG

160

β-Actin -R

GGAGCTTCCATTCCAATCAA

78257-F

TAGCGAGACAATCCTCCGTC

168

78257-R

TCGAGTCTGGGTTAGCTTGG

12139-F

CTTGGGGTCGCTTTTATCGG

175

12139-R

CACAGCCGTTAAAACACCCA

00457-F

GGCTCCTCCTCCTCTTAACC

194

00457-R

TCGACGCGACCATAGAATCA

42139-F

TTGGGGTTTTAACGGCTGTG

160

42139-R

AGCTCTCTCACCAACACCTC

27167-F

CTGACTACCGGTTACTCCCC

199

27167-R

CCGGACTGCCAAAAGATAGC

09899-F

TGCGGTGTCCTACATCAACT

181

09899-R

AATTGAAGAGGGAGGGACGG

6..67 line (Yuan et al): I think the references here should be shown in different way: keep the consequences in ref style

Response: Thank you for your important comments! We had rephrased This sentence as: “After analyzing the whole genome of Limonium bicolor, Yuan found the important genes affecting the formation of salt glands and the salt tolerance adaptation mechanism.”

7.Please provide clear goal of your study at the end of introduction section.

Response: Thank you for your important comments! We had improved the last paragraph of the introduction and proposed the novelty of this article. The content of the article is as follows: “As an important agricultural and animal husbandry area in northern China, Inner Mongolia has been experiencing an annual increase in saline alkali land area due to its arid climate and the fact that irrigation is the main planting method. Therefore, there is an urgent need to study salt tolerant mesophytes that are suitable for growing in the ecological environment of the region, in order to provide genetic resources for cultivating salt tolerant and highly resistant agricultural and pastoral crops. Based on the previous analysis of the morphology, physiology and Transcriptome of Glaux maritima, this study analyzed the salt-tolerance-related TFs of the salt-secreting halophyte Glaux maritima at the transcription level and identified six important genes distributed in the EIN, bHLH, and bZIP TF families. Clustering and expression analyses of these genes revealed that they play key roles in the response to high-salt conditions and could provide valuable genetic resources for cultivating salt-tolerant crops.”

Results

8.Figure 1. Please provide more details in the description of this figure. Figure 2、Figure 3、Figure 4 and Figure 7 Low quality. Need improvement. Captions for all figure should be better explain and descript.

Response: Thank you for your important comments! We had made modifications to each of the image questions cited in the article and added annotations to improve self-explanatory. Revised as follows:

Figure 1. Pie chart showing the percent share of identified transcription factors in each Tran-scription factor family. Note: The figure shows the distribution of 1466 transcription factor genes in the top 8 transcription factor families in the Glaux maritima. The 8 transcription factor families were ERF (133, 9.07%), bHLH (123, 8.39%), MYB related (95, 6.48%), NAC (94, 6.41%), WRKY (84, 5.73%), C2H2 (81, 5.53%), bZIP (70, 4.77%), and MYB (66, 4.50%).

Figure 2. KEGG pathway enrichment analysis of the transcription factor genes. Note: under salt stress, a total of 95 TF genes were successfully annotated in the KEGG database for Glaux maritima transcription factor genes. Among the three KEGG pathways, Plant hormone signaling pathway (66 genes), MAPK signaling pathway -plant (34 genes) and plant pathogen interaction (18 genes), the number of genes annotated is the largest.

Figure 3. GO enrichment analysis of the transcription factor genes. Note: under salt stress, A total of 54 functions were annotated, and 1191 genes were enriched in GO annotation and enrichment analysis. Among the three ontological processes, bindng (GO:0005488) and nuclear acid binding transcription factor activity (GO:0001071) had the highest distribution of Molecular Function; (GO:0005623), cell part (GO:0044464), and organelle (GO:0043226) had the highest distribution of Cell Component; cellular process (GO:0009987), metabolic process (GO:008152), regulation of biological process (GO:0050789), and biological regulation (GO:0065007) had the highest distribution of Biological Process.

Figure 4. Gene expression module diagram for trend analysis of all transcription factors. Note: After trend analysis, there were a total of 820 differentially expressed genes distributed in 8 expression modules. They were profile0(83 genes), profile1(239 genes), profile2(60 genes), profile3(62 genes), profile4(28 genes), profile5(62 genes), profile6(222 genes), and profile7(64 genes). In addition, the gene expression in the four modules of profile 0, 1, 6, and 7 showed significant differences after different concentrations of salt stress (p < 0.05). N0:0 mM/L NaCl stress; N3:600mM/L NaCl stress; N4:800mM/L NaCl stress.

Figure 7. Heat map of differential transcription factor gene expression and log2 FPKM expression trends. Note: a: Heat map of 33 up-regulated differentially expressed genes under different salt concentrations; b: log2 FPKM expression trends of 33 upregulated differentially expressed genes under different salt concentrations; c: Heat map of 35 down-regulated differentially expressed genes under different salt concentrations; d: log2 FPKM expression trends of 35 down-regulated differentially expressed genes under different salt concentrations. N0:0 mM/L NaCl stress; N3:600mM/L NaCl stress; N4:800mM/L NaCl stress.

9.Lines 242-245 、Lines 258-260、Lines 275-278– this text should be transfer to the Discussion section

Response: Thank you for your important comments! We had changed the three expressions mentioned by the evaluation experts to be discussed. In the fourth and fifth paragraphs of the discussion, respectively. The two paragraphs after rewriting are as follows: “As the first discovered gaseous plant hormone, ethylene is involved in the com-plete processes of plant growth and development and in various stress responses. The ethylene signaling pathway has also been clearly described. Ethylene insensitive 3/ethylene insensitive 3-like (EIN3/EIL) is a family of small TFs in plants mainly in-volved in ethylene signal transduction [33]. Some studies have shown that overexpres-sion of EIN3/EIL TFs in Arabidopsis and grapes can enhance salt resistance [34–36]. As an important family of nuclear TFs in the ethylene signaling pathway, EIN3/EIL has at-tracted extensive research attention. The genes in this small transcription factor family play a crucial role in the salt tolerance and stress resistance of plants [15,35,36,55–58]. In Arabidopsis, as the core component of ethylene signal transduction, EIN2 mutants are very sensitive to salt stress. CEND of EIN2 attenuates the salt hypersensitivity pheno-type of the mutant. The EIN3 mutant responds normally to salt stress, while the EIN3/EIL1 double mutant has a phenotype similar to that of the EIN2 mutant and is extremely sensitive to high-salt conditions [59]. EIN3 and EIL1 regulate the ethylene response in seedlings, and seedlings have higher salt tolerance than do mature plants [15,16]. In this study, 9 genes of the EIN3/EIL TF family were found enriched in the ethylene signaling pathway and may be related to salt tolerance in G. maritima. Ac-cording to changes in gene expression after salt stress, three significantly up-regulated DEGs were discovered. The above results extend understanding of ethylene signaling pathway regulation in response to salt stress and help explain the function of EIN3/EIL genes in salt tolerance. However, PPI network analysis showed no interacting proteins, indicating that these three genes may be highly conserved. These results lay a theoret-ical foundation for the functional verification of regulatory elements of the ethylene signaling pathway.

In this study, in addition to finding the above DEGs coding for positive regulatory transcription factors related to the ABA and ethylene signaling pathways, we also found DEGs coding for negative regulators in the MYC TF family related to the jasmonate signaling pathway. MYC TFs are widely expressed in animals and plants and have multiple regulatory functions [60]. When the GhMYC4 transcription factor in upland cotton was overexpressed in Arabidopsis, its salt tolerance and cold resistance were sig-nificantly improved, enhancing resistance of the plants to high-salt and drought condi-tions [61]. Identification and expression analysis of MYC TFs in tea plants showed that the TFs were related to amino acid anabolism under jasmonate stress, and that CsMYC1, 3, 5, and 8 were significantly related to photosynthesis-related pathways. These results showed that under stress, jasmonate signaling may change the amino acid content and photosynthetic efficiency of tea plants through MYC family expression, thus improving adaptability under stress [62]. MYC2, a negative regulatory transcription factor found in Arabidopsis, mediates salt sensitivity by inhibiting seed germination and delaying root growth; its knockout improves seed germination and plant root growth to better cope with salt stress [63]. These results are similar to the expression results of the two MYC family DEGs found in G. maritima. We found that these two DEGs interact with many proteins (Figure 9b, c), indicating that MYC TFs may play a key role in salt tolerance and stress resistance in G. maritima. The above results indicate that DEGs related to the MYC family in G. maritima participate most in protein regulation and mediation, which lays a theoretical foundation for explaining the function of salt resistance genes in G. maritima. These results have improved the theoretical basis for our subsequent functional verifi-cation experiments.”

  1. Reference section – a mess

Response: Thank you for your important comments! We had checked the format of all references and standardized all formats. The format is modified as: “Lindemose, S.; O’Shea, C.; Jensen, M. K.; Skriver, K. Structure, function and networks of transcription factors involved in abiotic stress responses. Int J Mol Sci .2013,14:5842–5878. DOI:10.3390/ijms14035842

Round 2

Reviewer 3 Report

Dear Chief Editor,

Thank you for providing the opportunity to review the revised manuscript. The authors have addressed all comments and incorporated changes suggested by reviewers during the first round of revisions. The revised version of the manuscript is improved as expected. Based on these revisions, now this study is a suitable contribution to the Agriculture. I recommend the manuscript for publication.

Thank you

With best regards

Dear Chief Editor,

Thank you for providing the opportunity to review the revised manuscript. The authors have addressed all comments and incorporated changes suggested by reviewers during the first round of revisions. The revised version of the manuscript is improved as expected. Based on these revisions, now this study is a suitable contribution to the Agriculture. I recommend the manuscript for publication.

Thank you

With best regards

Author Response

Thanks!

We have studied all comments carefully and have explained to clarify many related issues and made an appropriate revision in the manuscript. We think that the response has given the clear explanations on the confusing issues and the revised manuscript has been improved as a result of the constructive advices.

Reviewer 4 Report

Thank You for addressing my remarks.

Author Response

(The authors gave the same response as above.)
